# Computational Fluid Dynamics Approaches for Analyzing Rupture and Growth of Intracranial Aneurysms: A Systematic Review

**DOI:** 10.3390/biomedicines13122914

**Published:** 2025-11-28

**Authors:** Vincenzo T. R. Loly, Arthur Cintra, Felipe Ramirez-Velandia, Christopher S. Ogilvy, Emmanuel O. Mensah, João de Sá Brasil Lima, Mariana P. Nucci, Carlos E. Baccin, Lionel F. Gamarra

**Affiliations:** 1Hospital Israelita Albert Einstein, São Paulo 05652-900, SP, Brazil; vincenzololy@hotmail.com (V.T.R.L.); cintrakp@gmail.com (A.C.); carlos.baccin@einstein.br (C.E.B.); 2Neurosurgical Service, Beth Israel Deaconess Medical Center, Harvard Medical School, Boston, MA 02215, USA; felipegolframirez@gmail.com (F.R.-V.); cogilvy@bidmc.harvard.edu (C.S.O.); emensah2@bwh.harvard.edu (E.O.M.); 3Department of Mechanical Engineering, Instituto Mauá de Tecnologia, São Caetano do Sul 09580-900, SP, Brazil; joao.brasil@maua.br; 4LIM44, Hospital das Clínicas da Faculdade Medicina da Universidade de São Paulo, São Paulo 05403-000, SP, Brazil; mariana.nucci@hc.fm.usp.br

**Keywords:** intracranial aneurysm, CFD, hemodynamics, morphological parameters, rupture risk, aneurysm growth

## Abstract

**Background/Objectives**: Hemodynamic stressors, including abnormal wall shear stress (low or high) or oscillatory shear index are recognized as contributors to the pathogenesis, growth, and rupture of intracranial aneurysms (IAs). Computational fluid dynamics (CFD) has therefore become an essential tool for their quantitative assessment. This systematic review aimed to identify the most frequently analyzed hemodynamic and morphological parameters in recent CFD studies and summarize the methodological strategies employed. **Methods**: A systematic review was conducted following the PRISMA guidelines, including original studies published between 2019 and 2024 in PubMed, Scopus, Web of Science, and Embase databases. Eligible studies applied CFD to human saccular aneurysms addressing rupture or growth. Exclusion criteria comprised stent-assisted treatments, idealized or phantom models, and non-human or in vitro analyses. Extracted data included study characteristics, CFD software, meshing and solver approaches, and reported parameters. **Results**: Thirty-five studies met the eligibility criteria. Commercial software predominated across the segmentation, meshing, and solver stages. The most frequently evaluated wall shear stress metrics were the oscillatory shear index (OSI, 91.43%), time-averaged wall shear stress (TAWSS, 71.43%), low shear area ratio (LSAR, 60.00%), normalized wall shear stress (NWSS, 51.43%), and relative residence time (RRT, 45.71%). Morphological parameters such as the aspect ratio (AR, 74.29%), size ratio (SR, 68.57%), and volume (42.86%), reflecting aneurysm shape and relative size, were the most consistently evaluated and demonstrated strong associations with rupture and growth. **Conclusions**: A core set of morphological and hemodynamic parameters (AR, SR, TAWSS, OSI, RRT, and LSAR) was consistently identified as potential discriminators for the rupture and growth of intracranial aneurysms. However, substantial methodological heterogeneity and the absence of unified standards hinder reproducibility and clinical translation. Future research must urgently standardize computational frameworks, parameter definitions, and boundary conditions to enhance the consistency, comparability, and clinical applicability of CFD in aneurysm risk assessment.

## 1. Introduction

Intracranial aneurysms (IAs), estimated to affect 3.2% of the general population, are of significant clinical interest due to their propensity to rupture [1]. Aneurysmal rupture remains a devastating neurological event, affecting nearly 30,000 individuals annually in the United States and accounting for 0.4–0.6% of all deaths [2]. Following a subarachnoid hemorrhage, the one-year mortality rate is estimated at 35.3%, with a substantial proportion of survivors experiencing significant and lasting disability (16%) and fewer than half (48.7%) achieving a full return to their baseline level of functionality [3,4].

Multiple patient-related factors, such as smoking history, uncontrolled blood pressure, and collagen disorders, and aneurysm-specific characteristics including size, location, and morphology, have been associated with an increased risk of rupture [5]. However, the dynamics of blood flow within the cerebral vasculature are increasingly recognized as pathogenic contributors to the development and progression of cerebral aneurysms [6]. With advances in computational power and medical imaging, it is now possible to virtually reconstruct patient-specific vascular geometries and simulate complex flow patterns within hours using 3D imaging data. Computational fluid dynamics (CFD) is thus a field that bridges experimental observation and theoretical modeling, offering potentially clinically relevant applications [7].

CFD merges the principles of fluid mechanics, numerical methods, and computer science to solve fluid flow fields and predict their behavior. In the setting of IAs, understanding of flow proves a valuable tool for calculating metrics linked to aneurysmal growth and rupture. Shearing forces, whether low or high, contribute to the development and progression of aneurysms, as they drive endothelial dysfunction, inflammation, and weakening of the aneurysm wall [8]. Hence, the majority of CFD simulations have explored the wall shear stress (tangential, frictional force directed into the aneurysmal wall) as the most important factor predicting aneurysm evolution over time [9].

The initial foundational approach for CFD simulations included rigid wall models and steady flow due to its shorter computational time and justification in large vessels, as originally suggested by Steinman et al. [10]. Since then, CFD has transitioned from simplifying assumptions to more sophisticated models, including pulsatile flow with the possibility of calculating the oscillatory shear index (OSI), time-averaged metrics, and even more accurate fluid structure interaction (FSI) simulations accounting for vessel wall displacement and changes in flow from wall deformation [11,12,13,14]. These advancements, though limited in number, hold promise for enhancing the accuracy of simulations and provide a closer approximation of physiologic blood flow.

Despite these advances, the literature remains methodologically heterogeneous. In meta-analytic investigations, low wall shear stress and OSI have been strongly linked to aneurysmal rupture [9,15,16]. However, limitations remain when pooling data from the literature due to the heterogeneity of the simulations and aneurysms analyzed, which may vary in location, size, and morphology. Similarly, there is a lack of standardization across studies, especially in terms of morphological and hemodynamic parameters, as well as differences in boundary conditions. As a result, studies often rely on average or adapted boundary conditions derived from the literature. This lack of consistency in approaches can lead to divergent interpretations of results, complicating the comparison of findings and hindering the systematic application of CFD. In this context, standardization is needed to minimize sources of error and uncertainty. Aneurysm behavior enhances the reliability of cross-study comparisons. In addition, a wide variety of parameters have been analyzed across studies, increasing the heterogeneity of the available information and making it difficult for new investigations to identify the key parameters that should be included. Although such diversity reflects a positive effort toward broader exploration, it also contributes to a more fragmented and less consistent body of evidence. Consequently, critical analysis and careful filtering are required to determine which variables are truly relevant, most frequently used, and most promising for future research.

Therefore, given this scenario of fragmented and dispersed information that may hinder the advancement of the topic in the literature, the present systematic review aimed to analyze the available CFD studies on intracranial aneurysms (IAs). The focus was placed on the technical aspects of the simulations, including mesh generation, numerical methods, morphological parameters, and hemodynamic metrics, with the goal of consolidating the existing information and providing a solid reference base for future research. This base is intended to guide researchers in defining methodological approaches, selecting relevant parameters and metrics, and identifying the most appropriate computational tools and best practices to adopt.

## 2. Materials and Methods

### 2.1. Search Strategy

This systematic review was conducted in accordance with the Preferred Reporting Items for Systematic Reviews and Meta-Analyses (PRISMA) guidelines [17]. It includes articles from the PubMed, Scopus, Web of Science, and Embase databases, covering a five-year period (prior to 2024). The selection criteria were based on the following keyword sequences: (Intracranial Aneurysm OR Cerebral Aneurysm OR Brain Aneurysm) AND (Computational Fluid Dynamics OR CFD)):

PubMed: (((((“Computational Fluid Dynamics”[Title/Abstract]) OR (CFD[Title/Abstract])) AND (((“Intracranial Aneurysm”[Title/Abstract]) OR (“Cerebral Aneurysm”[Title/Abstract])) OR (“Brain Aneurysm”[Title/Abstract]))) AND (English[Language])) NOT (review[Publication Type])) AND ((“2019/01/01”[Date - Publication] : “2024/08/14”[Date - Publication])).

Scopus: ((TITLE-ABS-KEY (“Intracranial Aneurysm”) OR TITLE-ABS-KEY (“Cerebral Aneurysm”) OR TITLE-ABS-KEY (“Brain Aneurysm”))) AND ((TITLE-ABS-KEY(“Computational Fluid Dynamics”) OR TITLE-ABS-KEY (cfd))) AND PUBYEAR >2018 AND PUBYEAR<2025 AND (EXCLUDE(DOCTYPE,“re”) OR EXCLUDE(DOCTYPE,“ch”) OR EXCLUDE(DOCTYPE,“le”) OR EXCLUDE (DOCTYPE,“cp”)) AND (LIMIT-TO(LANGUAGE,“English”)).

Web of Science: (((TS=(“Intracranial Aneurysm”) OR TS=(“Cerebral Aneurysm”) OR TS=(“Brain Aneurysm”))) AND (TS=(“Computational Fluid Dynamics”) OR TS=(CFD))) AND (LA=(English)) AND (PY=(2019-2024)) NOT (DT=(“Review”) OR DT=(“Chapter”) OR DT=(“Letter”) OR DT=(“Conference Paper”)).

Embase: (‘intracranial aneurysm’:ti,ab,kw OR ‘cerebral aneurysm’:ti,ab,kw OR ‘brain aneurysm’:ti,ab,kw) AND (‘computational fluid dynamics’:ti,ab,kw OR cfd:ti,ab,kw) AND [english]/lim AND [2019-2024]/py NOT ((review:pt OR letter:pt OR conference:pt) AND abstract:pt OR book:pt).

This systematic review has been prospectively registered in PROSPERO (International Prospective Register of Systematic Reviews) under the identification number CRD420251145060.

### 2.2. Inclusion Criteria

For this review, we included only original research articles published in English between 2019 and 2024 (a five-year interval). The inclusion criteria were also defined based on the PICO framework: (P) Problem consisted of models of cerebral saccular aneurysms. (I) Intervention involved the application of CFD to assess the ruptured risk and growth of aneurysms. (C) Comparison included studies analyzing different hemodynamic parameters or aneurysm characteristics, such as morphological and biomechanical aspects. (O) Outcome focused on the prediction of aneurysm growth or rupture risk based on computational modeling. These criteria ensured the selection of studies that align with the objectives of this systematic review, emphasizing the use of CFD in the biomechanical analysis of cerebral aneurysm progression.

### 2.3. Exclusion Criteria

The following exclusion criteria were adopted: (i) analysis in aneurysm treated with Stent/Flow Diverter/Web, (ii) ideal model, (iii) porous zone/porous media modeling, (iv) porous media, (v) phantom or not human, (vi) in vitro studies only, (vii) non-saccular aneurysms, (viii) fenestration, (ix) recanalization, (x) patients with age < 18, (xi) studies not about risk of rupture or growth analysis, (xii) not CFD analyses, (xiii) review articles, (xiv) case reports, (xv) publications in languages other than English, and (xvi) indexed articles published in multiple databases (duplicates).

### 2.4. Data Extraction

The analysis of the reviewed papers focused on four key aspects, which were presented in tables covering the following: (1) aneurysm study characteristics and demographic data; (2) software utilization at different stages of CFD analysis; (3) numerical simulations, including preprocessing (mesh) and processing (solver) setup; (4) parameters encompassing morphological and hemodynamic factors; (5) wall shear stress-related parameters; (6) low shear area ratio criteria.

### 2.5. Risk of Bias Assessment

This study employed a peer-review approach in which the authors were independently and randomly assigned to evaluate the articles. The predefined inclusion and exclusion criteria guided the selection process. In cases of disagreement, a third author was consulted to make the final decision on whether to include or exclude the article.

### 2.6. Methodological Quality Assessment

The methodological quality of the studies included after the systematic screening process was assessed using the ROBINS-I tool [18]; it was applied in an adapted form due to the specific characteristics of the studies included in the systematic review (engineering analyses based on computer simulations), which do not correspond to conventional study designs in the health sciences. Among the available risk-of-bias assessment tools, **ROBINS-I** was chosen because it allows greater flexibility for adaptation across most domains, thereby minimizing the number of “not applicable” items during the assessment process. Accordingly, the following domains were evaluated: bias due to confounding, bias in selection of participants into the study, bias due to missing data, bias in measurement of outcomes, and bias in selection of the reported result. The domains bias in classification of interventions and bias due to deviations from intended interventions were not applied, as they were not relevant to the types of studies included. Each domain was rated according to standardized criteria and classified as “low”, “moderate”, “serious”, “critical”, or “no information”. The assessment followed a peer-review approach, in which three authors (V.T.R.L., A.P.C., and F.R.V.) independently evaluated the ROBINS-I domains. In cases of disagreement, a fourth author (L.F.G.) resolved the assessment. Graphical summaries of the risk-of-bias evaluations were produced with the robvis package in R, supported by its Shiny-based interface web application [19].

### 2.7. Data Compilation

In this review, all authors (V.T.R.L., A.P.C., F.R.V., J.B.L., C.S.O., C.E.B., E.O.M., M.P.N., and L.F.G.) independently and randomly analyzed, reviewed, and assessed the eligibility of titles and abstracts in pairs, following the established search strategy. The authors V.T.R.L., A.P.C., C.E.B., and L.F.G. selected the final articles by evaluating the full texts that met the selection criteria.

V.T.R.L. and A.P.C. were responsible for the initial screening of the selected articles, verifying their relevance based on the predefined inclusion and exclusion criteria. F.R.V. and E.O.M. contributed by extracting data on study characteristics, including the types of CFD analyses performed. M.P.N. and C.E.B. focused on evaluating the incorporation of morphological and hemodynamic parameters in the studies, ensuring consistency across the dataset. L.F.G. supervised the compilation and categorization of CFD software usage, numerical simulation settings, and key methodological aspects.

All authors participated in the review and synthesis of the data, ensuring a comprehensive and structured analysis of the selected studies.

### 2.8. Data Analysis

The data extracted from the selected studies were systematically analyzed using percentage distribution and categorical classification to identify key trends, variations, and potential outliers. The analysis focused on morphological and hemodynamic parameters relevant to aneurysm rupture and growth risk assessment. The results were organized into tables to facilitate comparison across different methodologies, while key findings were also illustrated through selected figures to enhance data interpretation.

## 3. Results

### 3.1. Systematic Process for Article Selection Based on PRISMA Guidelines

A comprehensive literature search was conducted in the Scopus and PubMed databases, covering five years up to 2024. This process initially yielded a total of 1147 articles. During the screening phase, 163 articles from the PubMed database were excluded (162 due to duplication within the Scopus database and 1 review), and 35 articles were removed after assessing eligibility (1 incorporated porous zones, 4 studies involving stents, flow diverters, or WEB devices, 4 studies unrelated to CFD analysis, 11 studies with a sample size of n < 10, and 15 studies is not about risk of rupture or growth analysis).

Among the 437 articles retrieved from the Scopus database, 11 were removed during the screening phase due to ineligibility, comprising 9 review articles and 2 non-article documents (Erratum and Editorial). Following a detailed eligibility assessment, 391 additional articles were excluded for the following reasons: 2 focused exclusively in vitro models, 5 focused exclusively on phantom models, 5 addressed recanalization, 12 investigated non-saccular aneurysms, 18 were unrelated to CFD analysis, 46 had a sample size of N < 10, 21 utilized idealized models, 22 incorporated porous zones, 80 examined aneurysm treated with stent, flow diverter, or WEB, and 178 did not analyze the risk of aneurysm rupture or growth.

From the 284 records retrieved from the Web of Science, 1 corresponded to a book chapter and 2 were meeting abstracts, while the remaining 281 were duplicates of studies already identified in Scopus and PubMed. Similarly, of the 228 records retrieved from Embase, 2 were excluded as preprints, and the remaining 226 were duplicates of articles found in other databases.

After excluding all articles that did not meet the criteria, only 35 were approved [20,21,22,23,24,25,26,27,28,29,30,31,32,33,34,35,36,37,38,39,40,41,42,43,44,45,46,47,48,49,50,51,52,53,54], as described in the flowchart in Figure 1A. In Figure 1B, the distribution of the 35 included articles by geographical region and publication year is presented. China accounted for the highest proportion of studies (41.67%), followed by the United States with 25%. Most of the studies were published in 2022, with nine studies predominantly conducted in Asian countries.

### 3.2. Overview of Aneurysm Study Characteristics and Demographic Data

In the analysis of Table 1, which presents the characteristics of the study, the demographic information of the sample, and information on the aneurysms, we observed that, in the last five years, China has stood out in research on this topic, contributing 43% of the studies included in this review [20,22,25,31,32,34,35,36,38,40,41,45,48,51]. The United States follows with 26% [23,26,29,30,33,39,43,44,53], then Japan with 17% [21,24,27,37,47], the Netherlands with 6% [49,50], and lastly, with the smallest percentage, are Chile [54], Korea [42], and Greece [28], with 3% of the studies. Regarding the objectives of the studies, 66% focused on analysis of aneurysm rupture [22,24,25,28,29,30,31,32,33,35,36,38,40,41,42,45,46,47,48,50,52,53,54], while 34% concentrated on growth [20,21,23,26,27,34,37,39,43,44,49,51]. Additionally, only six studies (17%) [23,30,37,39,43,44] compared the influence of growth or rupture in aneurysms with blebs to those without blebs.

Of the 35 studies included in this review, 18 (51.4%) explicitly reported the morphological subtype of the aneurysms analyzed, as recorded in the “Aneurysm subtype” column. Among them, six studies [21,29,30,37,39,40] investigated both sidewall and bifurcation aneurysms, which are associated with moderate and high rupture risk, respectively. Four studies [32,34,39,41] analyzed irregular and bifurcation aneurysms together, both considered to have a high risk of rupture. The remaining studies focused on a single subtype: five studies [27,33,35,50,52] evaluated only bifurcation aneurysms, two [31,46] focused exclusively on aneurysms with an irregular morphology, and one [49] analyzed only sidewall aneurysms (Table 1). These subtypes were associated with rupture risks ranging from moderate to high, depending on the morphology evaluated.

Additionally, these subtypes were observed in both single and multiple configurations per patient, which directly influenced the number of aneurysms analyzed in each study. Overall, the studies that reported aneurysm subtypes typically evaluated more than 100 aneurysms, reinforcing the relevance of these analyses in the context of hemodynamic evaluation using CFD.

The IA size classification proposed by the UCAS Japan study [55,56]—categorizing aneurysms as small (<5 mm), medium (>5 mm to 10 mm), large (>10 mm to 25 mm), and giant (>25 mm)—was adopted as the reference framework for stratifying the data included in this review. Among the 35 studies analyzed, 12 (34.3%) did not report aneurysm size, making classification according to this criterion unfeasible. Of the remaining 23 studies (65.7%) that provided quantitative data, 7 [20,21,26,31,37,40,50] exclusively evaluated aneurysms classified as small, corresponding to 20.0% of the total. Eight studies [32,34,41,45,47,50,51], representing 22.9%, included aneurysms within the small and medium size categories. Another eight studies [23,25,28,29,30,35,42,45], also accounting for 22.9%, reported aneurysms spanning the small, medium, and large categories. Although no study explicitly classified aneurysms as giant, one study reported aneurysms reaching 26 mm in diameter [30], which technically falls within the definition of a giant aneurysm.

The distribution of intracranial aneurysm sites among the included studies was analyzed based on the anatomical classification of cerebral circulation, grouping the locations into anterior and posterior circulation. Within the anterior circulation, aneurysms of the middle cerebral artery (MCA) were the most frequently reported, being present in 23 studies (66%) [21,23,24,25,27,28,29,30,32,33,34,35,36,37,39,42,46,47,49,50,51,52,54]. The internal carotid artery (ICA) was reported in 21 studies (60%) [21,23,24,25,26,28,29,30,32,34,36,37,39,42,46,47,49,50,51,53,54], followed by the anterior cerebral artery (ACA) in 16 studies (46%) [21,24,28,29,30,31,33,34,36,37,39,40,50,51,53]. Aneurysms of the posterior communicating artery (PCOM) were reported in 15 studies (43%) [23,26,30,32,34,36,39,41,42,45,47,50,53,54], while the anterior communicating artery (ACOM) was cited in 11 studies (31%) [23,25,26,30,34,39,47,50,51,52].

Regarding the posterior circulation, aneurysms of the basilar artery (BA) were identified in 11 studies (31%) [21,24,30,33,37,39,47,50,53,54], and the vertebral artery (VA) was mentioned in 9 studies (26%) [21,24,25,36,37,39,47,51,53]. Only one study (3%) classified aneurysms generically as located in the posterior circulation (PC) [50], without anatomical detail. Finally, non-specified or less common locations were grouped under “other” and appeared in two studies (6%) [23,30].

Analysis of the “follow-up” variable across the included studies revealed substantial variation in the duration of clinical monitoring. Of the 35 studies analyzed, only 11 (31.4%) [20,21,24,26,27,37,46,47,50,51,53] clearly reported this parameter, with follow-up periods ranging from a few months to over 13 years. The longest follow-up durations were reported by Leemans et al., with a range of 0.5 to 13 years [49], and Fujimura et al., with a mean of 6.8 years [37]. Other studies also presented extended follow-up periods, such as Perera [47], with 86 months, and Kimura et al. [51], ranging from 44.7 to 76.3 months. Follow-up durations close to four years were found in Miyata et al. [27], with 48.5 months, and Detmer et al. [53], reporting 900.8 days for ruptured aneurysms and 2432.1 days for unruptured cases. Some studies reported only the mean follow-up time, such as Yan et al. [20], with 15.9 months, and Weiss et al. [26], which indicated only “>1 year”. Fujimura et al. [24] described an average of 154.0 days, with a wide variation (±192.5 days), while Yuan et al. [46] reported a follow-up duration of 3 months.

Regarding the demographic data, the sample size, the average age of the patients, the sex distribution, and the comorbidities that could influence the characteristics of the aneurysm were analyzed, as shown in Table 1. The number of participants in the study varied from 10 to 1472, while the average age of the patients ranged from 55 to 64 years, with a sex distribution that was predominantly higher among women. Concerning comorbidities, systemic arterial hypertension was the most frequently analyzed, present in 49% of the studies [20,21,24,26,27,31,32,34,35,36,37,40,41,45,46,47], followed by smoking habits at 40% [21,24,26,27,31,34,35,36,37,41,45,46,47,52]. Diabetes mellitus [21,26,27,31,34,41,45,46,47], alcohol consumption [24,27,32,34,35,36,41,45,47], and dyslipidemia [24,26,27,34,35,36,37,41,45] were mentioned in 26% of the studies, while 14% reported a history of hypertension [21,31,41,50,53], and 9% mentioned cardiovascular diseases [32,41,45], cerebral ischemia [21,41,45], and heart disease [35,36,40]. Finally, 3% of the studies referred to coronary artery disease (CAD).

### 3.3. Software Used in CFD Analysis

CFD analysis is conducted in distinct stages, encompassing geometry preparation, mesh generation, and simulation setup, culminating in computational processing. These stages are typically performed using different software tools. Therefore, this systematic review examines the distribution of software utilized at each stage, specifically: segmentation (slicer software), which involves the creation and preparation of the geometry; mesh generation (mesh software), referring to the discretization of the geometry; and solving (solver software), which encompasses the configuration of the numerical analysis and execution of the simulation. The percentage distributions (percentage of total usage) of the results shown in Table 2 are illustrated with graphs as shown in Figure 2A.

In addition to categorizing software by stage, they are grouped into four categories: commercial, open source, in-house, or not informed. Commercial software dominates across all stages, accounting for 56.41%, 48.57%, and 61.11% of the usage for slicer, mesh, and solver software, respectively. Regarding the choice of segmentation software, all available tools provide the necessary functionalities to generate the base geometry required for CFD analysis. The main differences lie in the learning curve and, naturally, in the cost of commercial licenses, which may offer additional features not found in free alternatives—either for other applications or to streamline the segmentation process. In essence, all of these tools are suitable and sufficient for generating geometries for CFD simulations.

For segmentation, the leading commercial options are MIMICS (Materialise, Leuven, Belgium) (25.64%) [20,21,22,25,31,34,38,42,52] and Geomagic Studio (Geomagic, Morrisville, North Carolina) (10.26%) [31,41,45,46], while the most notable open-source alternative is VMTK (10.26%) [26,29,49,53], as shown in Figure 2A.

In mesh generation, Ansys ICEM CFD (Ansys Inc, Canonsburg, Pennsylvania, US) [21,25,31,38,40,41,45,46,47,48] stands out among commercial solutions (28.57%). Meanwhile, the open-source options, TetGen [26,29] and OpenFOAM [35,36], which include built-in mesh generation capabilities share equal representation with in-house software [52,53], each accounting for 5.71%, as shown in Figure 2B. Regarding the choice of mesh generator, any software is capable of producing both tetrahedral and hexahedral meshes. The tool itself does not directly influence the simulation results; however, differences may arise in the workflow used to prepare the input geometry, as well as in the ease and speed of mesh generation, depending on the efficiency of the algorithms implemented in each software.

For processing (CFD solvers), Ansys CFX (Ansys Inc, Canonsburg, Pennsylvania, US) (27.78%) [21,24,31,40,41,45,46,47,48,52] and Ansys Fluent (Ansys Inc, Canonsburg, Pennsylvania, US) (19.44%) [20,22,25,29,38,50,54] are the most used commercial solutions. The open-source alternative OpenFOAM [32,35,36] holds 8.33%, while 19.44% of users rely on in-house software [23,30,39,43,44,53], as shown in Figure 2C. Concerning the choice of solver software, both commercial and open-source platforms can deliver reliable and accurate results, as their implemented numerical models have undergone extensive verification and validation [57] both by the developers and through comparative studies available in the literature for various flow conditions. Such validation constitutes a fundamental prerequisite for their scientific and engineering application. Variations may arise primarily in the learning curve, cost, minor numerical deviations, and computational efficiency, depending on the algorithms employed [58,59,60]. When using in-house software, it is also crucial to perform validations and verifications. If it is impossible to compare directly with experiments, cases can be benchmarked with an analytical solution or verified with already validated software.

### 3.4. CFD Analysis Stages

#### 3.4.1. Preprocessing (Mesh Generation)

Prior to configuring the physical and numerical parameters of a CFD simulation, the vascular geometry must be discretized through computational mesh generation, a critical preprocessing step that directly impacts simulation results’ accuracy. Table 3 summarizes the key characteristics of the meshes reported across the reviewed studies, including mesh structure (structured, unstructured, or hybrid), element type (Figure 3A), quality metrics, number of prism (boundary) layers implemented near vessel walls to capture near-wall velocity special variations, and whether mesh-independence studies were conducted to verify solution convergence and mitigate discretization-induced errors.

Among the selected studies in this review, 82.9% utilized unstructured meshes [20,21,23,25,26,27,29,30,32,33,36,40,41,43,44,45,46,47,48,49,50,51,52,53,54], while 14.29% did not specify the mesh type [22,31,35,38,39], and one study used a Cartesian structured mesh [51]. Regarding element distribution (Figure 3E), tetrahedral elements [20,21,23,24,25,26,28,29,30,32,33,34,36,40,41,42,43,44,45,46,47,48,49,50,52,53,54] (Figure 3C) were the most commonly used (77.14%), followed by hexa-dominant [27], hexahedral [51] (Figure 3B), and polyhedral [37] (Figure 3D) elements, each representing 2.86%. Additionally, 14.29% of the studies did not report the element type used.

Prismatic element layers were also incorporated in some cases, as shown in Figure 3F. Most studies (51.43%) did not specify the number of layers. Among those that did, 14.29% used four layers [33,40,45,46,47,48], while the least common cases were seven layers [24] and zero layers [54], each accounting for 2.86%.

To assess mesh quality, various metrics are typically employed. However, none of the reviewed studies reported general element quality or aspect ratio metrics. Additionally, only 22.86% [23,26,28,29,30,42,52,54] of the studies conducted a mesh-independent analysis.

For novice CFD users, Figure 3 may serve as a useful reference for selecting an initial meshing approach and determining the preliminary number of prism layers to test. However, it is essential to perform a mesh-independence study and to ensure that mesh quality metrics comply with established best practices, rather than relying on the raw number of mesh elements or prism layers as definitive criteria. Each geometry presents its own specific characteristics, and adhering to these procedures is fundamental to maintaining scientific transparency and ensuring the reliability of the analysis.

#### 3.4.2. Physical and Numerical Simulation Setup

After mesh generation, the next step involves the physical and numerical setup. The physical setup consists of defining the flow state, boundary conditions, and fluid properties, including density and the chosen rheological model. The numerical setup, in turn, refers to the temporal discretization, characterized by the selected time-step size and the number of simulated cardiac cycles. The results corresponding to this stage are summarized in Table 4 and illustrated in Figure 4. All studies included in this systematic review modeled the flow as laminar, incompressible, and transient, employing a pulsatile inlet boundary condition to represent the blood flow rate at each instant of the cardiac cycle. Regarding the fluid setup, the blood viscosity model varied among studies: 94.29% adopted a Newtonian model [20,21,22,23,24,25,26,27,28,29,30,31,32,33,34,35,36,37,38,39,40,41,42,43,44,45,46,47,48,49,50,52,53], while 5.71% used the Casson model [51,54], a non-Newtonian approach. The distribution of viscosity coefficient values is illustrated in Figure 4A, with 40% of studies using 4.00 mPa·s (the highest reported value) and 14.29% using 3.45 mPa·s (the lowest viscosity value).

Another key physical property is density; however, since the flow is considered incompressible, there is no variation in density. As a result, the mass conservation and momentum equations are simplified, excluding density as a factor, meaning it does not influence the simulation outputs. Consequently, the percentage distribution presented in this review serves as a reference for readers when calculating mass flow, where applicable, and as a guideline for the range of values used to represent blood. The percentage distribution of density is shown in Figure 4B, and it can be seen that the two most commonly used values were 1060 [20,26,32,35,36,38,40,48] and 1050 kg/m^3^ [27,31,41,45,46,52], representing 22.86% and 17.14%, respectively.

For the transient aspect, the key parameters include time-step size and the number of cardiac cycles simulated, as shown in Figure 4C and Figure 4D, respectively. A time-step size of 1.0 × 10^−3^ s [25,29,31,41,45,46,48] was employed in 22.86%, representing approximately one-thousandth of the total duration of the phenomenon, given that a single cardiac cycle lasts less than one second. Additionally, 14.29% of studies used a time-step size of 1.0 × 10^−2^ s [22,30,38,44,50]. As for the number of cardiac cycles, 80% are distributed between two [22,23,24,30,31,37,38,39,43,44,47,49,53,54] and three [20,25,28,32,33,34,35,36,40,41,45,46,48,50] cycles, with each accounting for 40%. The remaining 20% consists of 11.4% not informed [26,27,51,52], 5.71% with four cycles [29,42], and 2.86% with only one cycle [21].

### 3.5. Parameters Analyzed in Aneurysm CFD Simulation

#### 3.5.1. Morphological Parameters

This systematic review does not consider basic one-dimensional measurements such as diameters (neck, maximum diameter, and vessel), heights, and angles due to variations in definitions and nomenclature. Instead, we focus on parameters derived from these fundamental measurements. 

Table 5 provides the definitions, Figure 5A illustrates these measurements (except angles), establishing them with geometric and mathematical rigor [62,63], while Table 6 summarizes all formulations for each parameter presented.

The morphological parameters were categorized into zero-order and second-order parameters, representing 95% [20,21,22,24,25,27,28,29,31,32,34,35,38,39,40,41,42,43,44,45,46,47,48,50,52,53,54] and 5% [39,43,44,50,53], respectively (Figure 5B). The zero-order parameters are classified between 2D shape (60%) and 3D shape (40%).

Table 7 summarizes all the parameters, ranking them by the number of uses, while Figure 5B presents these results graphically. Among the five most used (six parameters with a tie in fifth place), three parameters are from the 2D shape group, with AR, SR, and BF being first (16%), second (15.58%), and fifth (7.79%) most used, while for the 3D shape parameters, the most prominent are volume, NSI, and UI, being in third (9.74%), fourth (9.09%), and fifth positions (7.79%). Only 22.86% of the articles [23,26,30,33,36,37,50,51] did not analyze any morphological parameter.

#### 3.5.2. Hemodynamic Parameters

The hemodynamic parameter distribution is graphically illustrated in Figure 6. Additionally, Table 8 presents parameters with a frequency of use greater than three, while Table 9 provides a detailed breakdown of wall shear stress metrics, including average, maximum, minimum, and median values.

Among the five most frequently used hemodynamic parameters, all of which are related to wall shear stress, the most commonly evaluated was the OSI [20,21,22,23,24,25,26,28,29,30,31,32,33,34,35,36,37,38,39,40,41,42,43,44,45,46,47,48,50,52,53,54], reported in 91.43% of the studies included in this systematic review. This was followed by time-averaged wall shear stress (TAWSS) [20,21,22,23,25,26,27,28,30,32,33,34,35,36,38,39,41,42,43,44,47,50,52,53,54], reported in 71.43% of the studies. The third most frequently analyzed parameter was the low shear area ratio (LSAR) [21,22,24,26,29,30,31,33,34,35,36,38,39,40,41,42,44,46,48,50,53], appearing in 60% of the papers, followed by normalized wall shear stress (NWSS, normalized by the mean TAWSS of the parent artery) [21,24,30,31,32,34,35,36,37,39,40,41,44,45,46,48,52,53], reported in 51.43% of the studies. In fifth place was relative residence time (RRT) [20,21,23,25,28,31,34,35,36,40,41,42,43,45,47,54], reported in 45.71% of the papers.

High oscillatory shear area (HOA) [22,38], pressure loss coefficient (PLc) [33,37], gradient oscillatory number (GON) [37,52], energy loss coefficient [24,38], and normalized wall shear stress divergence (NWSSD, normalized by dynamic pressure) [24,37] were each reported in two studies, representing 5.71% of the studies analyzed.

Meanwhile, streamwise WSSG [35], spectral bandedness index (SBI), spectral power index (SPI) [33], time-averaged cross-flow index (TACFI) [28], aneurysmal inflow rate coefficient (AIRC) [27], total volume ratio (TVR) [42], aneurysm formation index (AFI), wall shear stress in systole and diastole [54], particle residence time (PRT) [50], volume flow rate [24], normalized transverse WSS (NtransWSS) [52], flow concentration ratio (FCR), high shear concentration ratio (HSCR), low WSS (LWSS), and high WSS (HWSS) [21] were each evaluated in one study, representing 2.86% of the studies included.

In contrast to LSAR, the high shear area ratio (HSAR), which represents the area where WSS exceeds a given threshold, was not included in the table due to its limited use, appearing in only one study. In that case, the threshold was set at WSS values surpassing 110% of the mean WSS. The LSAR parameters, which relate to the distribution of WSS on the surface of the aneurysm, are defined according to the thresholds selected in the studies and detailed in Table 10.

### 3.6. Clinical Results

#### 3.6.1. Aneurysm Growth and Bleb Formation

As for bleb formation, studies indicate that these structures develop preferentially in larger aneurysms with irregular morphology, elongated shape (higher aspect ratio), high OSI, heterogeneous WSS distribution, and the presence of concentrated blood inflow jets impacting the wall [30,39,43,44]. Distinctions between types of blebs have also been reported: aneurysms with thinner walls tend to be located near the inflow region of the aneurysmal sac, exhibiting high and heterogeneous WSS values, whereas aneurysms situated farther from the inflow region typically have thicker, often atherosclerotic walls and more uniform areas of low WSS [43]. Conversely, Karnam et al. [23] reported that regions of low WSS combined with high OSI are particularly conducive to wall weakening and bleb formation. This finding is not necessarily contradictory to studies reporting an association between bleb formation and high WSS, as these mechanisms may be complementary. In other words, both high and low WSS can be considered deviations from the physiological range of shear stress acting on the vessel wall. However, there is still no clear consensus in the literature as to whether both conditions contribute to bleb formation, or whether one predominates and, if so, whether it is the low- or high-WSS regime, or even other triggers acting together or being more relevant.

Regarding the overall growth of cerebral aneurysms, studies show that regions of low WSS, when associated with higher LSAR values, contribute to aneurysm progression. Yan et al. [20] reported that this effect is more pronounced when low WSS coincides with high OSI and RRT values, whereas Weiss et al. [26] highlighted low WSS alone as a significant factor. Although Leemans et al. [49] did not find strong predictive associations for growth, they observed that aneurysms which had enlarged over time exhibited lower WSS and higher LSAR values post-growth. In contrast, Kimura et al. [51] identified that concentrated regions of high WSS, surrounded by areas of low WSS during the peak systolic phase, may also promote growth. Additional studies have identified various predictors for aneurysm stability or progression, including RRT, SR, and irregularity [34], parameters such as WSSD and PLc for growth initiation [37], HSCR [21], and aneurysm-specific indices such as AIRC for those located in the middle cerebral artery [27].

#### 3.6.2. Aneurysm Rupture

Among the articles analyzed, three studies (8.57%) applied machine-learning techniques to improve rupture risk prediction, combining hemodynamic and morphological parameters in models based on the PointNet algorithm to extract hemodynamic cloud features [22,38] or using metrics derived from velocity informatics [29]. In line with these ML-based classification approaches, the study by Detmer et al. [53] highlights that population-specific differences can influence the predictive value of these parameters for rupture risk, thereby reinforcing the importance of developing models tailored to local clinical demographics. Among the hemodynamic parameters, high OSI was most frequently associated with increased risk, cited in six studies (17.14%) [28,30,35,40,41,42,47], followed by WSS also in six studies (17.14%) [25,28,36,42,46,52], with five (14.29%) detailing that low WSS is related to increased risk of rupture [25,28,42,46,52], elevated LSAR in four articles (11.43%) [36,41,42,48], and RRT in three (8.57%) [25,42,47]. Less frequently, NWSS was reported in two articles (5.71%), one analyzed in mirror aneurysm in the posterior communicating artery [41] and the other specifically in segment A1 [31]. One of the findings was that there is a significant reduction in NWSS after rupture [24]. Less frequently used parameters also appeared occasionally with correlation to the risk of rupture reported once (2.86%) each, namely the WSSG ratio, CHP [36], DWSS (WSS in diastole) [54], and new parameters introduced as TVR [42] and SPI [33]. According to Fukuda et al. [52], AAI may be useful for predicting the rupture risk of aneurysms, and PRT [50], a refined approach to RRT, did not show a relevant association with the risk of rupture.

Regarding morphological parameters, the SR emerged as the most frequently reported risk factor for aneurysm rupture, identified in six studies (17.14%) [31,32,41,46,47,54]. This was followed by the AR, reported in three studies [41,42,46], and aneurysm volume, indicated in two studies (5.71%) [24,47]. Additional parameters, such as the UI, NI [46], and BF [42], were each independently associated with rupture in one study each (2.86%). Furthermore, three studies (8.57%) examined ruptured aneurysms with blebs, suggesting that the presence of these protrusions may serve as a potential marker of increased rupture risk [28,30,32].

### 3.7. Methodological Quality Assessment

The distribution of risk of bias across the seven domains defined by the ROBINS-I tool is presented in Figure 7. D1 (bias due to confounding) showed moderate risk in the majority of studies (77.14%) [20,21,22,23,24,25,27,28,29,30,31,33,34,37,38,39,40,42,43,44,46,47,48,49,50,51,54], mainly reflecting heterogeneity in clinical and demographic characteristics of the populations analyzed. Similarly, D5 (bias due to missing data) presented moderate concerns in 94.28% of studies [20,21,22,23,24,25,26,27,28,29,30,31,32,33,34,35,36,37,38,39,40,41,42,43,44,45,46,47,48,49,50,51,52,54], particularly related to absent information on methodological quality, such as mesh-independence studies or detailed descriptions of numerical validation procedures.

By contrast, D2 (bias in selection of participants into the study), D6 (bias in measurement of outcomes), and Domain 7 (bias in selection of the reported result) were predominantly judged as low risk across most studies. D3 (bias in classification of interventions) and 4 (bias due to deviations from intended interventions) were not assessed, as they are not applicable to the computational fluid dynamics studies included in this review.

Together, these findings highlight that the main sources of potential bias in CFD-based aneurysm studies arise from baseline heterogeneity (D1) and insufficient methodological reporting (D5), while outcome-related domains are generally robust.

## 4. Discussion

When analyzing the distribution of aneurysm study characteristics, it is important to note the predominance of studies conducted in China, followed by the United States. This concentration of research in specific regions may reflect not only population size and research investment but also possible regional differences in clinical practice and reporting. Regarding demographic aspects, the consistent predominance of female participants across regions suggests a robust pattern rather than a study bias, reinforcing findings from the previous literature [1,64,65,66]. In addition, the average age range of participants (55–64 years) aligns with the literature on prevalence, supporting the notion that this age group represents a critical window for aneurysm occurrence [1].

Regarding comorbidities, hypertension was the most frequent (49% of studies), followed by smoking. This finding deserves attention, since hypertension is not only prevalent but also explicitly included in the PHASES score for predicting the 5-year rupture risk of intracranial aneurysms [67]. Nevertheless, the presence of such comorbidities within study groups does not automatically translate into an influence on the simulation outcomes. This limitation arises because their physiological consequences, such as blood viscosity alterations in diabetes or hemodynamic changes from hypertension, are usually excluded from computational models. The main reason for this omission is the restricted availability of patient-specific data, particularly in retrospective studies, and the logistical, ethical, and bureaucratic barriers faced in prospective designs. An important avenue for future research would be to integrate these clinical variables into mathematical models whenever such data are accessible, potentially allowing for more realistic simulations and better applicability to patient cohorts with specific comorbidities.

As for aneurysm characteristics, in terms of location, the three most frequently reported sites in the reviewed studies were MCA, ICA, and ACA, aligning with the distribution patterns consistently described in the broader literature [1]. These locations are also incorporated into the PHASES score; each is assigned a specific weighting based on its relative contribution to rupture risk [67]. In terms of size, most studies concentrated on small and medium aneurysms, while large aneurysms were also evaluated with notable frequency. In contrast, giant aneurysms were identified in only one study [30], representing a single case; this is expected, given that giant aneurysms represent an average incidence of 5% for all intracranial aneurysms [68]. To enhance the assessment of morphological parameters, future studies should ensure adequate stratification of aneurysm groups, whether by location, volume, or other relevant morphological metrics. An important aspect is the variability in follow-up duration. Although it does not directly influence the computational simulation of an aneurysm, it significantly impacts study design. The lack of consistent longitudinal follow-up hinders the detection of evolutionary patterns and prognostic factors, thereby limiting the scope and robustness of analyses and conclusions, particularly in identifying critical conditions or parameters associated with rupture risk.

A CFD analysis involves several stages, each requiring specialized software, and the best choice is the one that meets your needs most cost-effectively, considering factors such as price, learning curve, and technical support. In the studies reviewed, commercial software was the most widely used across all stages, while open-source tools appeared as the second most common option, particularly for segmentation and mesh generation. Nonetheless, this prevalence does not imply superiority over other software and should not be interpreted as a recommendation for adopting any specific tool.

Due to the complex and non-parametric nature of aneurysm geometry, mesh generation predominantly relies on tetrahedral or polyhedral elements. This expectation is corroborated by the results presented in Figure 3E and further supported by the studies summarized in Table 3. Most studies used an unstructured mesh, mainly with tetrahedral elements. The work that used a Cartesian mesh [51] did not respect the limits of the geometry; however, this approach was justified as the study was focusing on implementing the Lattice Boltzmann method, where one of the limitations is stepped geometries when dealing with complex fluid domains. Using hexahedral elements is impractical, as it is neither possible to generate a structured mesh (to generate a mesh that respects the contour of the geometry) nor to maintain a minimum quality criterion, such as skewness (<0.95) or orthogonal quality (>0.15) [61].

Regarding mesh quality, beyond the metrics mentioned before (including aspect ratio and others), it is essential to report mesh statistics and provide details of the mesh convergence study. Such information not only ensures transparency but also enables the assessment of mesh adequacy for the study’s validity, specifically the reliability of the results with respect to numerical errors (excluding model errors and physical simplifications). However, in the reviewed literature, none of the studies reported mesh quality metrics, and only 22.86% mentioned performing a mesh-independence study [23,26,28,29,30,42,52,54]. The details of the mesh independence study can be provided in supplementary material or even in the main text of the articles describing the procedure for tracking variations in relevant parameters from coarse to refined meshes, or by applying the Grid Convergence Index (GCI) method, a well-established approach in the literature [69].

Not performing a mesh-independence study constitutes a methodological limitation that compromises both the transparency and reliability of the reported results. In computational simulations, several sources of uncertainty may arise, including simplifications of the physical phenomena, discretization of the governing equations, boundary condition approximations, and numerical errors. Among these, numerical errors have been a subject of investigation over the decades [69,70], as they can be systematically reduced through mesh-independence analyses.

Numerical errors typically involve three main aspects: spatial convergence, temporal convergence, and iterative (or interaction) convergence. In transient analyses such as those involving aneurysmal hemodynamics, conducting independence studies is particularly critical. This involves refining the mesh to ensure spatial convergence, adjusting the time-step size to achieve temporal convergence, and testing the number of iterations to verify iterative convergence [71,72].

By performing these procedures, numerical errors are minimized, and the results become more robust and reliable, as further refinements no longer lead to significant changes in outcomes. Conversely, studies that omit mesh-independence testing may present results that vary substantially with mesh resolution or time-step selection, thereby increasing uncertainties and potentially compromising both the accuracy and the validity of the conclusions [71]. Therefore, with only 22.86% presenting mesh-independence studies, the goal for future studies is to routinely include these evaluations, along with consistent reporting of mesh quality metrics, as recommended in the fundamental guidelines for any CFD analysis. Most hemodynamic parameters are functions of wall shear stress, which is computed as the product of the viscosity coefficient and the velocity gradient in the near-wall region. Accurate estimation of this gradient is critical and is typically achieved by discretizing the mesh with prism layers. However, in complex geometries such as aneurysms, achieving this accuracy presents challenges, as it requires balancing mesh quality with the number and distribution of prism layers. Most articles did not provide information on the number of prism layers, but 14.29% used four prism layers, making it the most representative choice. The lower representation of higher layer counts is likely due to the increasing difficulty in maintaining good mesh quality as the number of layers increases. Only one case (2.86%) did not use prism layers [54], which is generally avoided, as tetrahedral elements are not suitable for this purpose [61]. It is important to note that in all studies, laminar flow was modeled, so the y+ parameter is not considered and is only used for turbulent flows. Even in the absence of the Y+ parameter, sufficient resolution in the near-wall region remains essential for accurately capturing velocity gradients. In addition to the number of prism layers, factors such as their total height, spacing, and growth factor are also relevant [61]. Thus, although most studies employed four layers and some extended this to seven, it is not possible to assert that these configurations provide adequate resolution. Once again, a mesh-independence study is necessary to address this question.

In the physical and numerical configuration of the simulation, a key consideration is the appropriate modeling of blood rheology. Blood is a multiphase viscoelastic fluid composed of a liquid phase (plasma) that behaves as a Newtonian viscous liquid and a dispersed transported phase consisting of red blood cells, platelets, and white blood cells [73]. Due to the high volume fraction of red blood cells, the overall mixture exhibits non-Newtonian behavior, primarily driven by interactions between the dispersed and continuous phases (when the red blood cell concentration exceeds 10% [74]). This complex behavior can be effectively described as a liquefied suspension of elastic cells, where viscosity varies dynamically based on shear rate, cell aggregation, and deformation [75,76]. In all reviewed studies, a common simplification in the simulations was to model blood as a single-phase fluid, representing its viscosity through either Newtonian or non-Newtonian models. According to the results of the systematic review, most of the studies adopted a Newtonian model, thereby neglecting viscosity variations with shear rate. The literature has yet to reach a definitive consensus on the most appropriate rheological model for blood, as its behavior can vary significantly depending on physiological conditions and flow characteristics. Even when using the Newtonian model, the choice of dynamic viscosity coefficient significantly impacts the results. In a hypothetical comparison of laminar flow in a pipe (Hagen–Poiseuille flow [77]), adopting a viscosity of 4.00 mPa·s (most used) results in 14.29% higher shear stress compared to a viscosity of 3.50 mPa·s (the second most used). Although adopting the same rheological model facilitates comparison of hemodynamic parameters, such as in studies employing the Newtonian model, it also introduces a modeling error due to the deviation from the actual non-Newtonian behavior of blood. This simplification can affect the interpretation of results in relation to the underlying pathology, potentially masking the true relevance of a parameter as a discriminator or, conversely, suggesting a false association.

Another key physical property is density; however, since the flow is considered incompressible, there is no variation in density. As a result, the mass conservation and momentum equations are simplified, excluding density as a factor, meaning it does not influence the simulation outputs [77,78]. Consequently, the blood density percentage distribution (Figure 4B) presented in this work serves as a reference for readers when calculating mass flow, where applicable, and as a guideline for the range of values used to represent blood.

In transient numerical simulations, the time-step size and the number of cardiac cycles simulated are critical parameters. The time-step size affects the discretization of physics over time, directly influencing the ability to capture high-frequency phenomena [61,79,80]. A time-step size that is too large can cause numerical instability, leading to divergence or inaccurate results [61,81]. Conversely, a time-step that is excessively small, beyond what is necessary, can drastically increase computational cost without meaningful improvements in accuracy [78]. Therefore, selecting an optimal time-step size is essential to balance accuracy and computational cost in transient simulations. The two most used time-step sizes were 1.0 × 10^−3^ [25,29,31,32,34,41,45,46,48] and 1.0 × 10^−2^ s [22,30,38,44,50], which both ensure a Courant number below 1, indicating that a time-step size in this range is enough to obtain plausible results. Beyond this, the time-step size must also be fine enough to discretize the waveform of the pulsatile inlet condition accurately. Most studies used two [22,23,24,30,31,37,38,39,43,44,47,49,53,54] or three cycles [20,25,28,32,33,34,35,36,40,41,45,46,48,50]; however, since most studies did not report performing a mesh-independence analysis or assess cardiac and temporal cycle independence (time-step sensitivity), it is not possible to conclude that two or three simulated cycles are sufficient to ensure reliable results. In addition to mesh independence, it is also necessary to verify the variation in parameters as a function of the number of iterations. The greater the number of cardiac cycles simulated, the more iterations are required for the solution to converge, that is, for the results to show no significant variations beyond a certain threshold. Therefore, there is no fixed or universal number of cycles to be adopted, as the number required for convergence depends on factors such as the mesh resolution and the selected time-step size.

Even when mesh-, time-step-, and iteration-independence analyses are performed, these procedures primarily ensure a reduction in numerical errors and increase the computational reliability of the results. However, they do not guarantee that the outcomes accurately represent the physical phenomenon, as modeling errors may still persist [71]. Addressing this type of error requires validation through experimental data, which may be unfeasible due to limitations in laboratory facilities, administrative constraints, or the lack of experimental infrastructure in health-related research particularly within engineering-focused institutions. Nonetheless, several studies have developed experimental validation methods, manufacturing phantoms of simulated cases and employing techniques such as Particle Image Velocimetry (PIV) [82] and Laser Doppler Spectroscopy (LDS) [83] to compare velocity and circulation fields, demonstrating that CFD can produce accurate and reliable results. Therefore, when combined with careful modeling and the minimization of numerical errors through proper independence analyses, following established best practices in computational simulation, the CFD approach can be confidently applied to the study of blood flow in cerebral aneurysms.

In a line of experimental work combined with numerical modeling, Bordás et al. [83] compared Newtonian and non-Newtonian fluids and demonstrated that Newtonian models tend to underestimate wall shear stress (WSS) and the oscillatory shear index (OSI) in most regions of the aneurysmal sac. They also reported inaccuracies in identifying high-OSI areas on the sac surface, which may mislead the hemodynamic interpretation of cerebral aneurysm pathophysiology. Beyond phantom-based validations, the Aneurysm-on-a-Chip model developed by Vivas et al. [84] integrated endothelial cell layers into microfluidic systems to investigate shear-induced biological responses. Similarly, the experimental works of Meng et al. [85] and Wong et al. [86] using chamber devices evaluated how different shear stress conditions affect endothelial morphology and function, revealing phenomena such as cell elongation, F-actin alignment, and altered expression of mechanosensitive genes (endothelial nitric oxide synthase—eNOS; nuclear factor kappa-B—NF-κB; and integrins). Collectively, these studies contribute to progressively bridging the gap between biological experimentation and CFD-based hemodynamic analysis, an approach that could support translation into clinical decision-making.

Another critical aspect concerns the impact of using generic versus patient-specific boundary conditions. CFD simulations of aneurysmal hemodynamics have shown that adopting generalized inflow profiles, instead of patient-specific ones, can lead to substantial discrepancies in wall shear stress (WSS) magnitudes. Moreover, aneurysms exhibiting variations in vortex structure and inflow jet dynamics were more frequently classified as hemodynamically unstable when generalized boundary conditions were applied. These findings highlight the importance of implementing patient-specific inflow conditions in CFD analyses to ensure accurate and physiologically representative hemodynamic calculations in cerebral aneurysms [87].

Most of the hemodynamic parameters that showed a correlation with the risk of rupture or growth are among the top five most used parameters, such as OSI, TAWSS, LSAR, NWSS, and RRT. This indicates that, in addition to having a set of parameters in common, these parameters have shown promise for clinical inferences. OSI and WSS (especially when low) stand out as discriminating factors for aneurysm growth and rupture, may contributing to wall weakening and eventual rupture [23,30,39,43,44]. Next, the LSAR and RRT parameters, which appeared as discriminants for both cases, give results consistent with low WSS, since the larger the region of low WSS, the greater the LSAR and RRT; therefore, the results are physically consistent [88,89]. The fact that these parameters emerged as discriminating factors with statistical significance is intrinsically linked to the specific study designs, boundary conditions, group stratification, and other methodological characteristics. Additional WSS-derived parameters, such as NWSS, CHP, and WSSG, also showed correlations with rupture risk. Novel or highly specific parameters, including SPI, [33] AAI [52], and TVR [42], stood out within the studies in which they were first introduced, showing potential for future investigation to clarify their predictive value for aneurysm rupture. Similarly, parameters such as WSSD, PLc [37], HSCR [21], and AIRC [27] were associated with aneurysm growth, suggesting possible roles as discriminators; however, their true relevance should be assessed in more heterogeneous aneurysm cohorts that encompass a broader range of sizes and anatomical locations.

Studies aiming to predict aneurysm rupture risk rely directly on the identification of effective discriminants; therefore, refining which parameters are most relevant is crucial for developing more accurate prediction models. As noted earlier, determining whether a parameter truly functions as a discriminant, and thus contributes meaningfully to rupture risk, requires careful stratification along with consideration of local, clinical, and demographic factors. This point is reinforced by Detmer et al. [53], who demonstrated that population-specific differences can significantly influence the predictive value of these parameters. The same principle applies to future investigations on aneurysm growth prediction, where parameter relevance may likewise vary according to population characteristics and study context.

Morphological parameters were more frequently identified as significant or discriminating factors in rupture studies [31,32,35,41,42,46,47]. Both growth and rupture analyses consistently highlighted AR and SR, two shape-related parameters, whose common one-dimensional measure is the height (Table 6). For consistency across studies, it is essential that diameter calculations maintain geometric coherence; in this context, using the hydraulic diameter [62] is recommended. Additionally, parameters such as UI, NI [46], and BF [42] were identified as potential discriminants for rupture risk. With respect to bleb formation, evidence suggests that larger and more irregular aneurysms, particularly those with higher AR, are more prone to developing blebs [44]. The presence of blebs itself also emerges as a potential discriminating factor for rupture risk [28].

Other less frequently used parameters, such as second-order curvature metrics (e.g., GAA, GLN, MAA, and MLN) and additional shape descriptors (e.g., IPR, VOR, and CR), although not associated with clinical inferences in the reviewed studies, warrant further investigation. Their potential as discriminators could be better assessed in analyses involving more structured aneurysm cohorts, both larger and more heterogeneous, as well as in demographically localized or site-specific studies, thereby enabling a deeper understanding of their relevance in rupture risk assessment. However, it is necessary to standardize nomenclatures and mathematical definitions (respecting geometric fundamentals) to ensure the reproducibility of results and consistency across different articles.

For the literature to converge in identifying truly discriminatory parameters, it is essential to standardize nomenclature, calculation methods, and even boundary conditions (except when studies aim to examine specific scenarios, given that most employed pulsatile conditions are modeled without patient-specific data due to ethical or bureaucratic constraints).

Regarding the sources of bias in the included articles, one of the main contributors is the heterogeneity of the populations analyzed (D1). Studies with larger samples often combine data from different centers and patient groups with diverse clinical and demographic characteristics. Conversely, studies with smaller samples, although sometimes stratified according to specific criteria, still present heterogeneity in other aspects. Both scenarios may introduce selection bias (D2), either through inclusion criteria based on clinical parameters or through the exclusion of cases due to missing information, poor image quality, and similar issues. Furthermore, a frequent limitation across most studies was the lack of detailed reporting on CFD mesh characteristics, particularly regarding mesh quality and convergence criteria.

The primary objective of this systematic review was to identify the general hemodynamic and morphological parameters applied in CFD studies of cerebral aneurysms. However, due to the exclusion of simulations involving aneurysms treated with stents, WEB devices, flow diverters, or porous media modeling, the analysis did not incorporate parameters specific to these treatment scenarios, which may nonetheless serve as important predictors of rupture or growth. In addition, the review was restricted to saccular aneurysms, which represent the most prevalent type, but this criterion excludes morphological parameters that may be relevant to other aneurysm subtypes, such as fusiform or fenestrated aneurysms. Another limitation arises from the heterogeneity of the included study designs, ranging from case–control studies and cohorts to analytical cross-sectional studies, each with distinct levels of bias. Although this variability does not directly compromise the ranking of parameters, it may influence the clinical interpretation of results.

## 5. Conclusions

Across studies conducted in different regions of the world, various methodological approaches and computational tools were employed. A consistent set of hemodynamic and morphological parameters was identified, all of which proved to be relevant discriminators for the rupture or growth of intracranial aneurysms. Among the hemodynamic parameters, those related to wall shear stress (TAWSS, NWSS, OSI, RRT, and LSAR) were the most frequently analyzed, with the area-weighted average being the predominant quantitative metric, highlighting the importance of shear stress in aneurysmal behavior. Regarding morphological parameters, AR and SR were the most reported, both describing geometric proportions of the aneurysm, while V emerged as a promising indicator of overall aneurysm size. Together, these metrics provide a more comprehensive description of aneurysm morphology than the basic measurements typically used in clinical practice, which generally include only neck diameter, sac size, and height. It is recommended that they be retained in future research, while also encouraging the investigation of additional metrics, such as TVR and AAI, which require evaluation in larger and more heterogeneous populations. In addition, it is necessary to follow best practices and conduct mesh-independence, iteration, and time-step size studies.

Furthermore, simulations should undergo validation to ensure accurate representation of reality and to assess whether Newtonian simplifications suffice or if a particular non-Newtonian model is more appropriate.

The development of deterministic models for rupture risk estimation, whether probabilistic or classification-based, will directly benefit from further refinement of these discriminants. Incorporating local clinical and demographic data is also crucial to improve model accuracy, as population-specific differences may influence the predictive value of these parameters for rupture risk. In this context, the complementary nature of different studies can help fill existing gaps in the medical literature, ultimately providing a stronger foundation for more robust and reliable conclusions, if they are properly standardized in the parameters analyzed and are based on clinical data.

A common feature across all studies is the reliance on physical and mathematical parameters as the foundation for analysis. However, the interpretation of results and the conclusions drawn from the same parameter may vary among studies due to intrinsic differences in modeling approaches, such as boundary conditions or parameter calculation methods. Therefore, it is essential to standardize the methodologies used for parameter computation to enable meaningful comparisons between studies. This need is particularly evident for morphological parameters, especially by establishing unified formulas for those based on diameter and height ratios, as well as for simple geometric measurements. Such standardization should employ the concept of hydraulic diameter and provide clear definitions for height measurements—whether orthogonal to the neck plane, along the maximum distance, or based on other consistent geometric criteria—rather than relying on simplified clinical definitions that are often geometrically inconsistent. Nonetheless, these simplifications remain necessary for physicians to make preliminary inferences in clinical practice while translational research and the integration of engineering and medicine continue to advance. In this regard, several publications, such as Ma et al. [62], focus specifically on the definition and calculation of morphological parameters and may serve as methodological references. Moreover, when patient-specific boundary conditions are unavailable, researchers should adopt standardized reference conditions or establish confidence intervals to account for interindividual variability. This approach allows for the estimation of comparable ranges while maintaining consistency, except in cases involving specific physiological alterations, such as exercise or pathological changes.

## Figures and Tables

**Figure 1 biomedicines-13-02914-f001:**
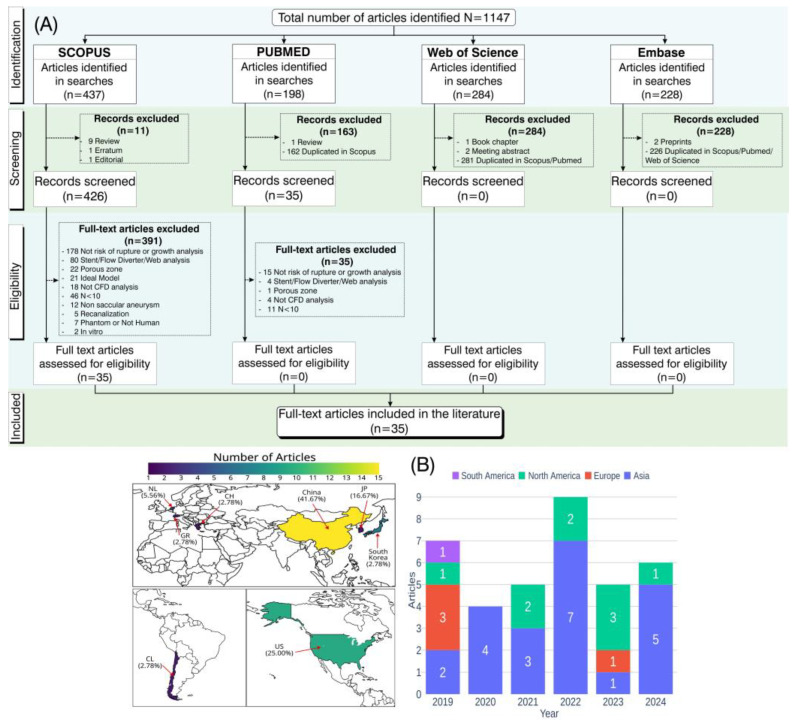
Schematic representation of the process for article identification, screening, and eligibility assessment in this systematic review, following PRISMA guidelines. (**A**) Flowchart summarizing the steps from identification to inclusion of articles. (**B**) Distribution of included articles by geographical region and publication year.

**Figure 2 biomedicines-13-02914-f002:**
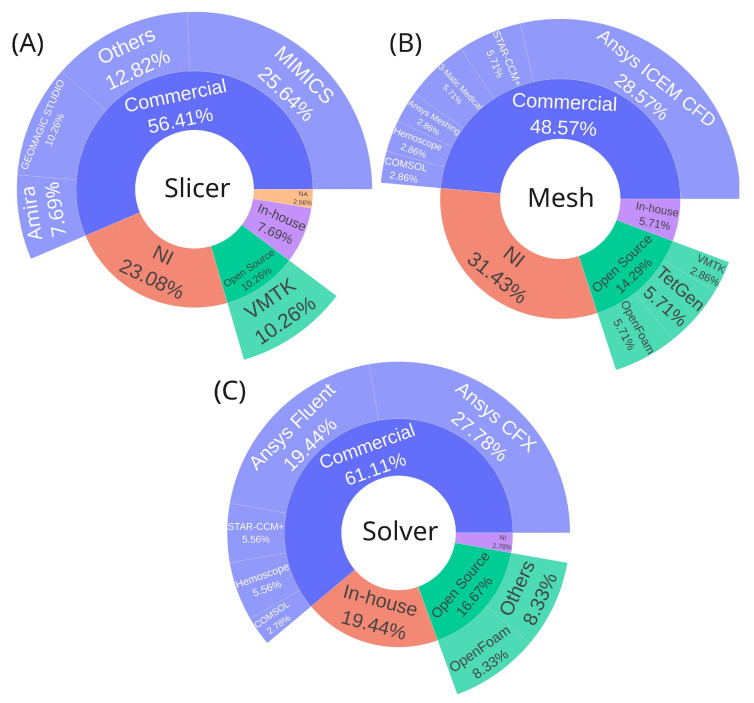
Software distribution across CFD analysis workflow stages. (**A**) Segmentation software for geometry creation and preparation from medical images. (**B**) Meshing software for geometry discretization into computational grids. (**C**) Solver software for numerical simulation configuration and execution. Software platforms were classified as commercial, open-source, in-house, or not informed. Abbreviations: NI, not informed; NA, not applicable.

**Figure 3 biomedicines-13-02914-f003:**
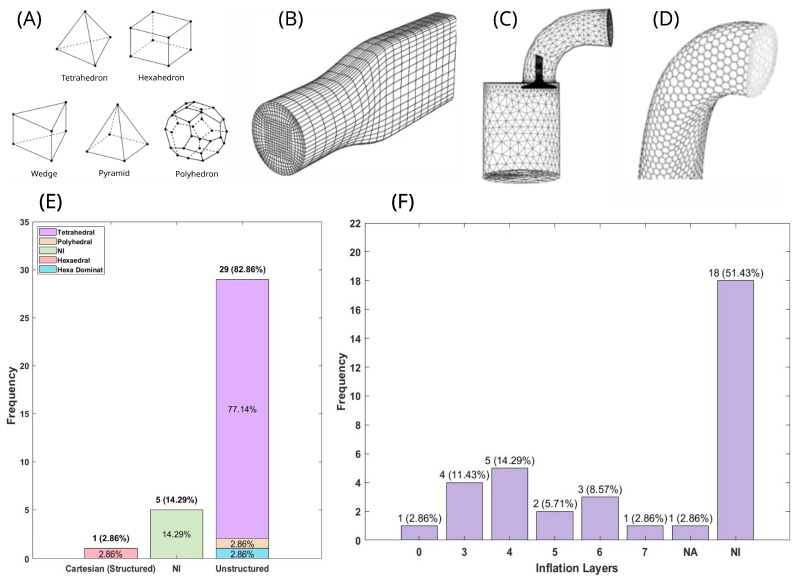
Computational mesh characteristics and distribution of meshing strategies in cerebrovascular CFD studies. (**A**) Common cell (element) types employed in mesh generation, illustrating the fundamental geometric building blocks for spatial discretization (**B**–**D**) Mesh structure examples: (**B**) structured mesh composed of hexahedral cells arranged in an organized grid pattern [61]; (**C**) unstructured mesh utilizing tetrahedral cells with flexible node connectivity [61]; (**D**) hybrid unstructured mesh combining multiple element types for geometric adaptability. (**E**,**F**) Statistical distribution of meshing approaches across reviewed studies: (**E**) percentage distribution of mesh structure types and predominant element geometries employed; (**F**) percentage distribution of boundary layer (inflation/prism layer) implementation strategies for near-wall flow resolution. Abbreviations: NI, not informed; NA, not applicable.

**Figure 4 biomedicines-13-02914-f004:**
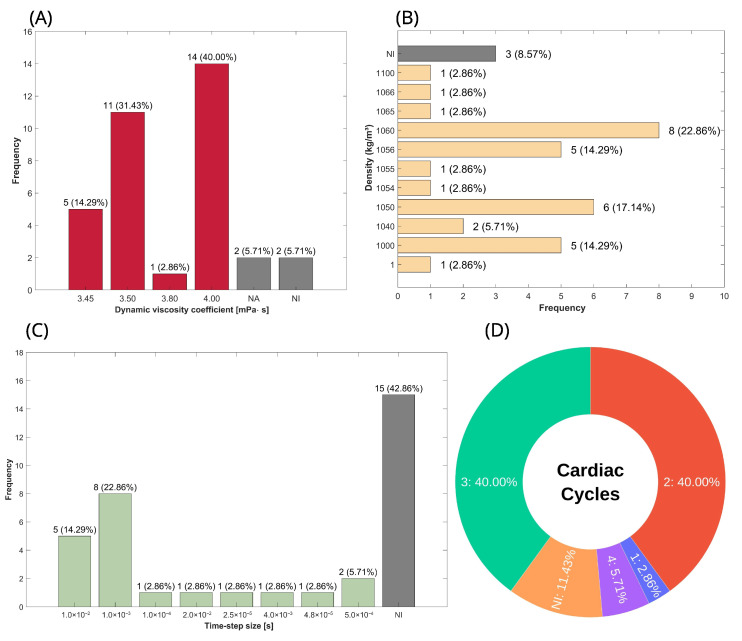
Distribution of blood physical properties and transient parameters used in the configuration of CFD simulations, characterizing both the viscous behavior of blood and the temporal discretization setup. (**A**) Percentage distribution of blood viscosity in studies using the Newtonian model; gray bars represent non-Newtonian or non-reported cases. (**B**) Percentage distribution of blood density. (**C**) Percentage distribution of the time-step size used for the temporal discretization of transient simulations. (**D**) Percentage distribution of the number of cardiac cycles considered for the development of simulations. Abbreviations—NI: not informed; NA: not applied.

**Figure 5 biomedicines-13-02914-f005:**
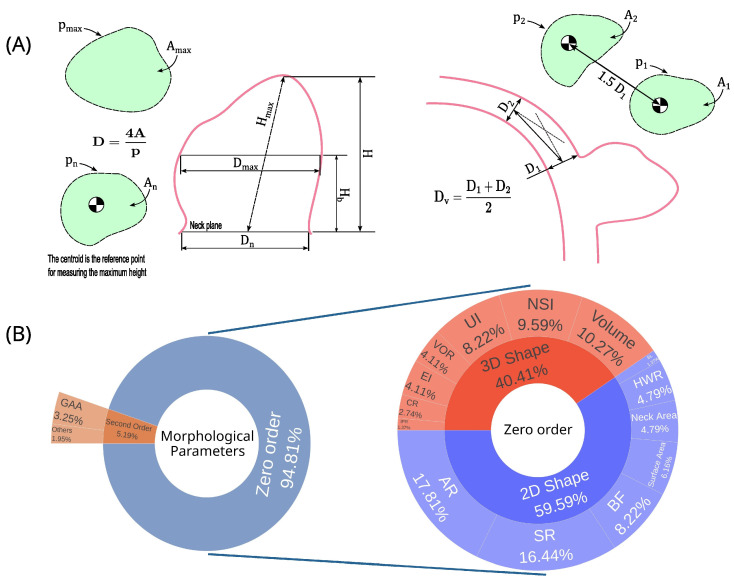
Aneurysm morphological parameters distribution and one-dimensional measurements. (**A**) Aneurysm one-dimensional parameters; (**B**) Percentual distribution of morphological parameters used in the papers analyzed in the systematic review. Abbreviations—A: cross-sectional; D: diameter; H: height; p: perimeter; aspect ratio; SR: size ratio; NSI: nonsphericity index; BF: bottleneck factor; UI: undulation index; HWR: height-to-width ratio; VOR: volume-to-ostium area ratio; EI: ellipticity index; GAA: area-averaged Gaussian curvature; CR: convexity ratio; IPR: isoperimetric ratio; BL: bulge location; GLN: L2-norm of Gaussian curvature; MAA: area-averaged mean curvature; MLN: L2-norm of mean curvature.

**Figure 6 biomedicines-13-02914-f006:**
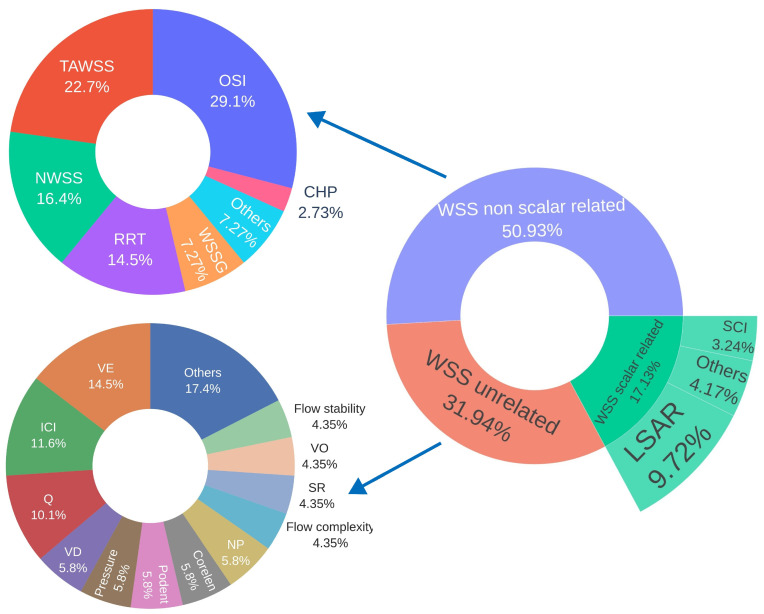
Percentage distribution of hemodynamic parameters classified as scalars and non-scalars related to wall shear stress, as well as unrelated scalars. Abbreviations—WSS: wall shear stress; OSI: oscillatory shear index; TAWSS: time-averaged wall shear stress; LSAR: low shear area ratio; NWSS: normalized wall shear stress; RRT: relative residence time; VE: mean velocity; WSSG: wall shear stress gradient; ICI: inflow concentration index; Q: mean inflow rate into aneurysm; SCI: shear concentration index; VD: mean viscous dissipation; Pressure: average aneurysm pressure; Podent: proper orthogonal decomposition entropy; Corelen: vortex core line length; NP: normalized pressure; SR: mean shear rate; CHP: combined hemodynamic parameter; nCrPoints: nr of WSS critical points; VO: mean vorticity.

**Figure 7 biomedicines-13-02914-f007:**
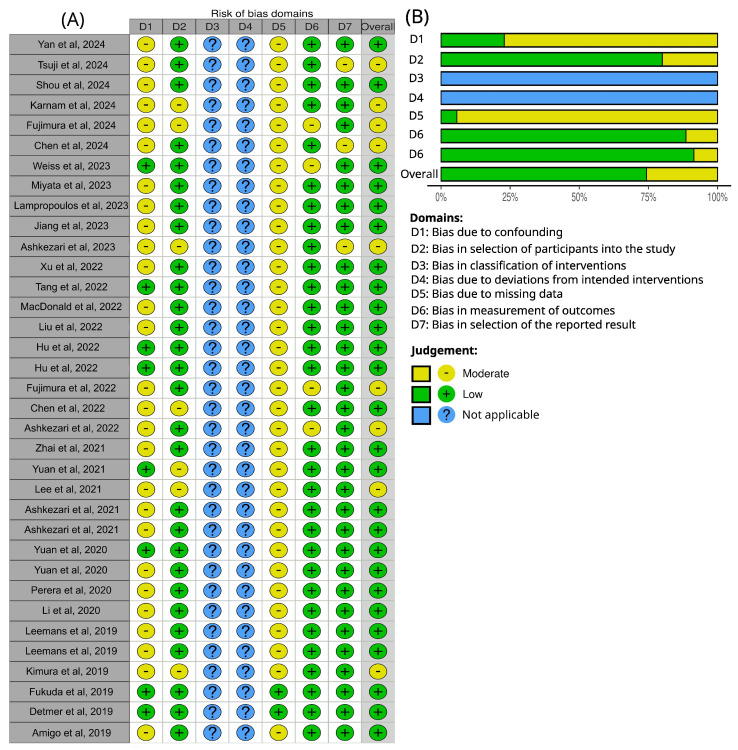
ROBINS-I risk of bias assessment for the articles analyzed in the systematic review [20,21,22,23,24,25,26,27,28,29,30,31,32,33,34,35,36,37,38,39,40,41,42,43,44,45,46,47,48,49,50,51,52,53,54]. (**A**) Traffic light plot of bias domains for each study. (**B**) Summarize of bias risk for each domain; both generated via robvis in shiny web interface [19].

**Table 1 biomedicines-13-02914-t001:** Descriptive analysis of aneurysm study characteristics and demographic data.

Study	Year	Country	Study Aneurysm Focus	Group Types (Aneurysm Samples)	Aneurysm Subtype	Aneurysm Number	Aneurysm Sample	Imaging Source	Aneurysm Size (mm)	Aneurysm Site	Follow-Up Mean/Duration (Range)	Patient Sample	Mean Age	Sex Distribution F:M	Concomitant Disease
Growth															
Yan et al. [20]	2024	China	Growth	Stable (66)/ Growing (30)	NI	NI	96	CTA, 3D TOF MRA	2–5	Intracranial	15.9 months (6–38.4 months)	96	NI	NI	HTN
Tsuji et al. [21]	2024	Japan	Growth	Stable (182)/ Growing (33)	70 sidewall, 145 bifurcation	Single (125), multiple (60)	215	CTA or DSA, MRA	3–5	ICA (102), MCA (59), ACA (35), VA/BA (19)	1170 d	185	64	134:51	91 HTN, 16 DM, 15 previous CI, 12 previous SAH, 23 smoking
Karnam et al. [23]	2024	USA	Growth	Aneurysm with bleb (150) Aneurysm without bleb (209)	-	Single (209), 150 with blebs (213)	359	3DRA or CTA	5–11	ICA (66), MCA (152), ACOM (68), PCOM (40), Other (33)	NI	268	NI	NI	NI
Weiss et al. [26]	2023	USA	Growth	Stable (11)/ Growing (11)	NI	NI	22	MRA, CTA	2.5–4.0	ICA (12), ACOM (4), PCOM (4), BA (2)	>1 y	22	63	18:4	8 HTN, 8 HLD, 3 DM, 10 smoking,
Miyata et al. [27]	2023	Japan	Growth	Stable (58)/ Growing (25)	M1-2 83 bifurcation	Single	83	DSA, CTA, or MRA	NI	MCA (83)	48.5 months	83	66.9	55:28	50 HTN, 23 HLD, 8 DM, 21 smoking, 15 alcohol
Liu et al. [34]	2022	China	Growth	Unruptured (272): Stable (182)/ Growing (49)	85 bifurcations, 49 irregular shapes	41 multiple IAs	272	CTA	3.4–6.7	ICA (187), MCA (62), ACOM/ACA (24)	NA	231	53.2	133:98	94 HTN, 20 HLD, 5 DM, 6 CAD, 5 iStroke, 32 smoking, 23 alcohol
Fujimura et al. [37]	2022	Japan	Growth	Stable (34)/ Growing (10)	13 sidewall, 19 bifurcation	NI	44	3D angiographic imaging	2.4–3.4	ICA (13), MCA (12), ACA (7), BA (6), VA (6)	6.8 yrs	20	64.6	13:7	8 HTN, 2 HLD, 6 smoking
Ashkezari et al. [39]	2022	USA	Growth	Aneurysm with bleb (739/22): Unruptured (375)/Ruptured (341)/Unknown (19) Aneurysm without bleb (1660/19): Unruptured (1304)/Ruptured (307) Unknown (49)	132 sidewall, 603 bifurcation 635 sidewall, 1025 bifurcation	452 single, 283 multiple 724 single, 936 multiple	2395/266	3DRA or CTA	NI	ICA (884), MCA (520), PCOM (403), ACOM (367), ACA (84), BA (128), VA (9)	NI	1614/195	57.0	1037:396 181 NI	NI
Ashkezari et al. [43]	2021	USA	Growth	Aneurysm with thin bleb (22), atherosclerotic bleb (19)	NI	NI	32	3DRA, CTA	NI	Dome (22) Body (17) Neck (2)	NI	NI	NI	NI	NI
Ashkezari et al. [44]	2021	USA	Growth	Rupture: Aneurysm with bleb (97) Aneurysm without bleb (173)	NI	NI	270	NI	NI	NI	NI	199	54.5	NI	NI
Leemans et al. [49]	2019	Netherlands	Growth	Stable (81)/ Growing (56)	51 sidewalls	NI	137	CTA, MRA, or 3DRA	3.5–7.6	ICA (55), MCA (35), PCOM (15), ACOM (16), ACA (6), PC (10)	0.5–13 yrs	137	57.3	101:18 18 NI	16 previous SAH
Kimura et al. [51]	2019	China	Growth	Stable (6)/ Growing (6)	NI	NI	12	TOF-MRA	2.1–10.1	ICA (5), MCA (2), ACOM (3), ACA (1), VA-BA (1)	44.7/76.3 months	12	68.3	10:2	NI
Rupture															
Shou et al. [22]	2024	China	Rupture	Unruptured (109)/Ruptured (40)	NI	NI	149	DSA	NI	NI	NA	149	NI	NI	NI
Fujimura et al. [24]	2024	Japan	Rupture	Unruptured (NI)/ Ruptured (NI)	NI	NI	21	CTA, DSA	~3–7	ICA (9), MCA (3), ACA (4), VA (2), BA (3)	154.0 ± 192.5 days	21	56.8 ± 15.1	13:8	9 HTN, 4 HLD, 7 smoking, 8 alcohol use
Chen et al. [25]	2024	China	Rupture	Unruptured (11)/Ruptured (106)	NI	NI	11	4D-CTA	~3–17	ICA (1), MCA (5), ACOM (1), PCOM (1), VA (3)	NA	10	62.2	6:4	NI
Lampropoulos et al. [28]	2023	Greece	Rupture	Unruptured (12)	NI	NI	12	NI	~3–16	ICA (10), MCA (10) ACA (1),	NA	12	NI	NI	NI
Jiang et al. [29]	2023	USA	Rupture	Unruptured (68)/Ruptured (44)	51 sidewall, 41 bifurcation	NI	112	MRA	4–25	ICA (39), MCA (52) ACA (21),	NA	112	NI	NI	NI
Ashkezari et al. [30]	2023	USA	Rupture	Rupture: Aneurysm with bleb (349) Aneurysm without bleb (324)	96 sidewall, 577 bifurcation	510 single, 163 multiple IAs	673	3DRA	1.5–26.3	ICA (82), MCA (123), ACA (29), PCOM (165), ACOM (203), BA (41) Other (30)	NA	615	54.5 (12–93)	372:190	NI
Xu et al. [31]	2022	China	Rupture	Unruptured (26)/Ruptured (23)	A1 segment, 11 irregular shapes	9 multiplicities	49	CTA	2.12–4.45	ACA (A1)	NA	49	60 (37–75)	33:16	26 HTN, 5 DM, 8 smoking, 3 previous SAH
Tang et al. [32]	2022	China	Rupture	Unruptured (52)/Ruptured (52)	47 irregular, 81 bifurcation	NI	104	DSA	~3–11	ICA (5), PCON (31), MCA (16)	NA	52	63.2	41:11	25 HTN, 5 CHD, 11 drinking
MacDonald [33]	2022	USA	Rupture	Unruptured (24)/Ruptured (26)	50 bifurcations	NI	50	NI	NI	ACA MCA BA	NA	50	NI	NI	NI
Hu et al. [35]	2022	China	Rupture	Unruptured (40)/Ruptured (40)	80 bifurcations	NI	80	DSA	2.9–10.8	MCA	NA	40	57.8 (41–77)	22:18	27 HTN, 14 HLD, 10 smoking, 9 drinking, 5 heart disease
Hu et al. [36]	2022	China	Rupture	Unruptured (91)/Ruptured (91)	NI	189 single	182	NI	NI	ICA (4), MCA (44), PCOM (40), ACA (1), VA (2),	NA	91	58.1	52:39	58 HTN, 32 HLD, 29 smoking, 20 drinking, 15 heart disease
Chen et al. [38]	2022	China	Rupture	Unruptured (109)/Ruptured (39)	NI	NI	148	3DRA	NI	NI	NA	NI	NI	NI	NI
Zhai et al. [40]	2021	China	Rupture	Unruptured (20)/Ruptured (12)	6 sidewall, 26 bifurcation	17 multiple	32	DSA	1.2–5.6	ACA (pericallosal artery)	NA	31	57.6	21:10	23 HTN, 3 heart disease
Yuan et al. [41]	2021	China	Rupture	Unruptured (72)/Ruptured (72)	69 irregular shape, 77 bifurcation	NI	144	DSA	3.25–8.15	PCOM	NA	72	58.2	57:15	29 HTN, 23 HLD, 10 DM, 15 atherosclerosis, 5 CI, 12 CHD, 6 smoking, 8 drinking, 7 previous SAH
Lee et al. [42]	2021	Korea	Rupture	Unruptured (11)/Ruptured (12)	NI	NI	23	3DRA, CTA	7.37–12.71	ICA (6), PCOM (17),	NA	23	62.8 (40–86)	NI	NI
Yuan et al. [45]	2020	China	Rupture	Unruptured (115)/Ruptured (83)	83 single ID, 115 bifurcations	83 single ID	198	DSA	1.81–5.99	PCOM	NA	198	58.7	141:57	110 HTN, 37 HLD, 18 DM, 30 smoking, 33 alcohol, 18 CI, 21 CHD
Yuan et al. [46]	2020	China	Rupture	Unruptured (12)/Ruptured (12)	13 irregular shapes	NI	24	3DRA	3.9–11.2	ICA (10), MCA (2)	3 months	12	50.7	7:5	7 HTN, 3 DM, 4 smoking
Perera [47]	2020	Japan	Rupture	Unruptured (38)/Ruptured (10)	NI	NI	48	TOF-MRA, PC-MRI	3–10	ICA (3), MCA (17), PCOM (10), ACOM (8), BA (9), VA (1)	86 months	48	NI	33:15	33 HTN, 4 DM, 8 smoking, 7 drinking
Li et al. [48]	2020	China	Rupture	Unruptured (133)/Ruptured (23)	NI	NI	156	NI	NI	NI	NA	148	NI	NI	NI
Leemans et al. [50]	2019	Netherlands	Rupture	Unruptured (61)/Ruptured (53)	80 bifurcations	NI	114	3DRA	4.3–9.1	ICA (20), MCA (35), ACA/PCOM/Posterior (59)	NA	137	55	NI	NI
Fukuda et al. [52]	2019	Japan	Rupture	NI	84 bifurcations	NI	84	CTA	3.6–7.68	ACOM (42) MCA (42)	NA	84	70.4	43:39	62 HTM, 49 smoking
Detmer et al. [53]	2019	USA	Rupture	Unruptured (1513)/Ruptured (616)	NI	Single (1727), multiple (402)	2129	3DRA	NI	ICA (749) MCA (469) ACA (70) ACOM (322) PCOM (344) VA (42) BA (133)	NA 900.8 days (ruptured), 2432.1 days (unruptured)	1472	57.08	1065:407	613 SAH
Amigo et al. [54]	2019	Chile	Rupture	Unruptured (32)/Ruptured (26)	NI	NI	58	3DRA	NI	ICA, MCA, PCOM, BA	NA	49	57.07 (combined)	NI	NI

Abbreviations: USA: United States of America; NI: not informed; ID: infundibular dilations; CTA: 3D computational tomography angiography; DSA: 3D digital subtraction angiography; MRA: magnetic resonance angiography; 3DRA: 3D rotational angiography; ICA: internal carotid artery; MCA: middle cerebral artery; ACA: anterior cerebral artery; VA: vertebral artery; BA: basilar artery; ACOM : anterior communicating artery; PCOM: posterior communicating artery; OS: ophthalmic segment; AchA: anterior choroidal artery; PICA: posterior inferior cerebellar artery; PC: posterior circulation; HTN: hypertension; DM: diabetes mellitus; CI: cerebral infarction; SAH: subarachnoid hemorrhage; HLD: hyperlipidemia; CHD: coronary heart disease.

**Table 2 biomedicines-13-02914-t002:** Summary of software used for segmentation, meshing, and numerical solving in CFD analyses.

Study	Segmentation		Mesh		Solver	
	Name	Version	Name	Version	Name	Version
Yan et al. [20]	MIMICS	21.0	3-Matic Medical	13.0	Ansys Fluent	19.0
Tsuji et al. [21]	MIMICS	24.0	Ansys ICEM CFD	2021 R2	Ansys CFX	2021 R2
Shou et al. [22]	MIMICS	13.0	NI	NA	Ansys Fluent	18.0
Karnam et al. [23]	NI	NA	NI	NA	In-House	NA
Fujimura et al. [24]	Amira	5.6	NI	NA	Ansys CFX	2020 R1
Chen et al. [25]	MIMICS	NI	Ansys ICEM CFD	2020 R2	Ansys Fluent	2020 R2
Weiss et al. [26]	VMTK	NI	TetGen	NI	SimVascular’s	NI
Miyata et al. [27]	Hemoscope	1.5	Hemoscope	1.5	Hemoscope	1.5
Lampropoulos et al. [28]	NI	NA	Ansys Meshing	NI	Nektar++	NI
Jiang et al. [29]	VMTK	NI	TetGen	1.4.2	Ansys Fluent	20.0
Ashkezari et al. [30]	NI	NA	NI	NA	In-House	NA
Xu et al. [31]	MIMICS/Geomagic Studio	17.0/9.0	Ansys ICEM CFD	11.0	Ansys CFX	11.0
Tang et al. [32]	Syngo Workplace	NI	3-Matic Medical	13.0	OpenFOAM	6.0
MacDonald [33]	NA	NA	VMTK	NI	Oasis	NI
Liu et al. [34]	MIMICS	17.0	STAR-CCM+	12	STAR-CCM+	12
Hu et al. [35]	Innova Workplace	NI	OpenFOAM	NI	OpenFOAM	NI
Hu et al. [36]	NI	NA	OpenFOAM	NI	OpenFOAM	NI
Fujimura et al. [37]	Amira	5.6	STAR-CCM+	10.06.010	STAR-CCM+	10.06.010
Chen et al. [38]	MIMICS	10.0	Ansys ICEM CFD	18.0	Ansys Fluent	18.0
Ashkezari et al. [39]	NI	NA	NI	NA	In-House	NA
Zhai et al. [40]	MIMICS	19.0	Ansys ICEM CFD	18.0	Ansys CFX	18.0
Yuan et al. [41]	Geomagic Studio	12.0	Ansys ICEM CFD	14.0	Ansys CFX	14.0
Lee et al. [42]	MIMICS	20.0	COMSOL	5.2	COMSOL	5.2
Ashkezari et al. [43]	NI	NA	NI	NA	In-House	NA
Ashkezari et al. [44]	NI	NA	NI	NA	In-House	NA
Yuan et al. [45]	Geomagic Studio	12.0	Ansys ICEM CFD	14.0	Ansys CFX	14.0
Yuan et al. [46]	Geomagic Studio	12.0	Ansys ICEM CFD	14.0	Ansys CFX	14.0
Perera [47]	Flova	NI	Ansys ICEM CFD	14.5	Ansys CFX	14.5
Li et al. [48]	In-House	NA	Ansys ICEM CFD	NI	Ansys CFX	14.0
Leemans et al. [49]	VMTK/In-House	NI/NA	NI	NA	NI	NA
Leemans et al. [50]	NI	NA	NI	NA	Ansys Fluent	NI
Kimura et al. [51]	OsiriX	NI	NI	NA	AN2WER/Hemoscope	NI/1.5
Fukuda et al. [52]	MIMICS	NI	In-House	NI	Ansys CFX	NI
Detmer et al. [53]	VMTK/In-House/Amira	NI/NI/5.6	In-House	NI	In-House	NI
Amigo et al. [54]	NI	NA	NI	NA	Ansys Fluent	NI

Abbreviations—NI: not informed; NA: not applied.

**Table 3 biomedicines-13-02914-t003:** Mesh characteristics regarding quality, element type, structure, and prism layers in CFD studies.

Ref.	Mesh Structure	Volumetric Element	Quality Metric	Number of Prism Layers	Max Aspect Ratio	Independence Study
Yan et al. [20]	Unstructured	Tetrahedral	NI	NI	NI	NI
Tsuji et al. [21]	Unstructured	Tetrahedral		6		NI
Shou et al. [22]	NI	NI		NI		NI
Karnam et al. [23]	Unstructured	Tetrahedral		NI		Yes
Fujimura et al. [24]	Unstructured	Tetrahedral		7		NI
Chen et al. [25]	Unstructured	Tetrahedral		5		NI
Weiss et al. [26]	Unstructured	Tetrahedral		NI		Yes
Miyata et al. [27]	Unstructured	Hexa-Dominant		3		NI
Lampropoulos et al. [28]	Unstructured	Tetrahedral		NI		Yes
Jiang et al. [29]	Unstructured	Tetrahedral		6		Yes
Ashkezari et al. [30]	Unstructured	Tetrahedral		NI		Yes
Xu et al. [31]	NI	NI		NI		NI
Tang et al. [32]	Unstructured	Tetrahedral		NI		NI
MacDonald [33]	Unstructured	Tetrahedral		4		NI
Liu et al. [34]	Unstructured	Tetrahedral		NI		NI
Hu et al. [35]	NI	NI		3		NI
Hu et al. [36]	Unstructured	Tetrahedral		3		NI
Fujimura et al. [37]	Unstructured	Polyhedral		6		NI
Chen et al. [38]	NI	NI		NI		NI
Ashkezari et al. [39]	NI	NI		NI		NI
Zhai et al. [40]	Unstructured	Tetrahedral		4		NI
Yuan et al. [41]	Unstructured	Tetrahedral		NI		NI
Lee et al. [42]	Unstructured	Tetrahedral		NI		Yes
Ashkezari et al. [43]	Unstructured	Tetrahedral		NI		NI
Ashkezari et al. [44]	Unstructured	Tetrahedral		NI		NI
Yuan et al. [45]	Unstructured	Tetrahedral		4		NI
Yuan et al. [46]	Unstructured	Tetrahedral		4		NI
Perera [47]	Unstructured	Tetrahedral		4		NI
Li et al. [48]	Unstructured	Tetrahedral		3		NI
Leemans et al. [49]	Unstructured	Tetrahedral		NI		NI
Leemans et al. [50]	Unstructured	Tetrahedral		NI		NI
Kimura et al. [51]	Cartesian (Structured)	Hexahedral		NA		NI
Fukuda et al. [52]	Unstructured	Tetrahedral		5		Yes
Detmer et al. [53]	Unstructured	Tetrahedral		NI		NI
Amigo et al. [54]	Unstructured	Tetrahedral		0		Yes

Abbreviations—NI: not informed; NA: not applied.

**Table 4 biomedicines-13-02914-t004:** Numerical simulation setups used in the selected articles, with respect to blood modeling, boundary condition specifications, and temporal discretization strategies.

Ref.	Flow State	Time-Step Size [s]	Inlet BC	Cardiac Cycles	Rheological Model	μ [Pa.s]	Density [kg/m^3^]
Yan et al. [20]	Transient	2.0 × 10^−2^	Pulsatile	3	Newtonian	3.50 × 10^−3^	1060
Tsuji et al. [21]	Transient	1.0 × 10^−4^	Pulsatile	1	Newtonian	3.50 × 10^−3^	1056
Shou et al. [22]	Transient	1.0 × 10^−2^	Pulsatile	2	Newtonian	4.00 × 10^−3^	1055
Karnam et al. [23]	Transient	NI	Pulsatile	2	Newtonian	4.00 × 10^−3^	1000
Fujimura et al. [24]	Transient	5.0 × 10^−4^	Pulsatile	2	Newtonian	3.50 × 10^−3^	1100
Chen et al. [25]	Transient	1.0 × 10^−3^	Pulsatile	3	Newtonian	3.50 × 10^−3^	1056
Weiss et al. [26]	Transient	NI	Pulsatile	NI	Newtonian	4.00 × 10^−3^	1060
Miyata et al. [27]	Transient	NI	Pulsatile	NI	Newtonian	4.00 × 10^−3^	1050
Lampropoulos et al. [28]	Transient	2.5 × 10^−5^	Pulsatile	3	Newtonian	3.45 × 10^−3^	1056
Jiang et al. [29]	Transient	1.0 × 10^−3^	Pulsatile	4	Newtonian	4.00 × 10^−3^	1040
Ashkezari et al. [30]	Transient	1.0 × 10^−2^	Pulsatile	2	Newtonian	NI	NI
Xu et al. [31]	Transient	1.0 × 10^−3^	Pulsatile	2	Newtonian	3.45 × 10^−3^	1050
Tang et al. [32]	Transient	4.0 × 10^−3^	Pulsatile	3	Newtonian	3.50 × 10^−3^	1060
MacDonald [33]	Transient	4.8 × 10^−5^	Pulsatile	3	Newtonian	3.50 × 10^−3^	1
Liu et al. [34]	Transient	1.0 × 10^−3^	Pulsatile	3	Newtonian	3.50 × 10^−3^	1056
Hu et al. [35]	Transient	NI	Pulsatile	3	Newtonian	4.00 × 10^−3^	1060
Hu et al. [36]	Transient	NI	Pulsatile	3	Newtonian	4.00 × 10^−3^	1060
Fujimura et al. [37]	Transient	NI	Pulsatile	2	Newtonian	3.50 × 10^−3^	1056
Chen et al. [38]	Transient	1.0 × 10^−2^	Pulsatile	2	Newtonian	4.00 × 10^−3^	1060
Ashkezari et al. [39]	Transient	NI	Pulsatile	2	Newtonian	NI	NI
Zhai et al. [40]	Transient	NI	Pulsatile	3	Newtonian	3.50 × 10^−3^	1060
Yuan et al. [41]	Transient	1.0 × 10^−3^	Pulsatile	3	Newtonian	3.45 × 10^−3^	1050
Lee et al. [42]	Transient	NI	Pulsatile	4	Newtonian	3.50 × 10^−3^	1066
Ashkezari et al. [43]	Transient	NI	Pulsatile	2	Newtonian	4.00 × 10^−3^	1000
Ashkezari et al. [44]	Transient	1.0 × 10^−2^	Pulsatile	2	Newtonian	4.00 × 10^−3^	1000
Yuan et al. [45]	Transient	1.0 × 10^−3^	Pulsatile	3	Newtonian	3.45 × 10^−3^	1050
Yuan et al. [46]	Transient	1.0 × 10^−3^	Pulsatile	3	Newtonian	3.45 × 10^−3^	1050
Perera [47]	Transient	NI	Pulsatile	2	Newtonian	3.80 × 10^−3^	1054
Li et al. [48]	Transient	1.0 × 10^−3^	Pulsatile	3	Newtonian	4.00 × 10^−3^	1060
Leemans et al. [49]	Transient	NI	Pulsatile	2	Newtonian	4.00 × 10^−3^	1000
Leemans et al. [50]	Transient	1.0 × 10^−2^	Pulsatile	3	Newtonian	4.00 × 10^−3^	1040
Kimura et al. [51]	Transient	NI	Pulsatile	NI	Casson	NA	NI
Fukuda et al. [52]	Transient	NI	Pulsatile	NI	Newtonian	3.50 × 10^−3^	1050
Detmer et al. [53]	Transient	NI	Pulsatile	2	Newtonian	4.00 × 10^−3^	1000
Amigo et al. [54]	Transient	5.0 × 10^−4^	Pulsatile	2	Casson	NA	1065

Abbreviations—NI: not informed; NA: not applied; BC: boundary condition; μ: viscosity coefficient for Newtonian fluid.

**Table 5 biomedicines-13-02914-t005:** Description of basic geometric measurements used in the calculation of morphological parameters.

Parameter	Name	Description
A	Area	Cross-sectional area
p	Perimeter	Cross-sectional perimeter
H	Height	Maximum perpendicular distance from the neck plane
$H_{b}$	---	Distance of the maximum diameter cross-section from the neck boundary plane
$D_{H}$	Hydraulic diameter	$D_{H}=4A/p$
$D_{n}$	Neck diameter	Aneurysm neck hydraulic diameter
$D_{max}\;or\;W$	Width/max diameter	Aneurysm width, that is, the maximum hydraulic diameter parallel to the neck
$D_{v}$	Parent vessel diameter	Hydraulic diameter of parent vessel
S	Aneurysm model surface area	Outer region of the aneurysm fluid domain/CAD model, equivalent to the tunica intima area within the aneurysm
$S_{ch}$	Convex hull area	Surface area of the convex hull volume
V	Aneurysm model volume	Volume occupied by blood within the aneurysm, i.e., volume of the aneurysm zone in the fluid domain
$V_{ch}$	Convex hull volume	Volume of the smallest fully convex surface that completely encloses the aneurysm sac, representing the minimum bounding volume of the aneurysm model

Abbreviations—A: cross-section area; p: cross-section perimeter.

**Table 6 biomedicines-13-02914-t006:** Description of morphological parameters, including names and mathematical formulations.

Parameter	Name	Calculation
AR	Aspect ratio	$AR=H/D_{n}$
SR	Size ratio	$SR=H/D_{v}$
NSI	Nonsphericity index	$NSI=1-{\left(18\pi \right)}^{1/3}\frac{V^{2/3}}{S}$
BF	Bottleneck factor	BF = $D_{max}/D_{n}$
UI	Undulation index	$UI=1-V/V_{ch}$
$A_{n}$	Neck (ostium) area	$A^{*}$
HWR	Height-to-width ratio	$HWR=H/D_{max}$
VOR	Volume-to-ostium ratio	$VOR=V/A_{n}$
EI	Ellipticity index	$EI=1-{\left(18\pi \right)}^{1/3}\frac{V_{ch}^{2/3}}{S_{ch}}$
GAA	Area-averaged Gaussian curvature	$GAA=\sum_{all\;triangles}K_{gi}S_{i}/\sum_{all\;triangles}S_{i}$
CR	Convexity ratio	$CR=V/V_{ch}$
IPR	Isoperimetric ratio	$IPR=S/V^{2/3}$
BL	Bulge location	$BL=H_{b}/H$
GLN	L2-norm of Gaussian curvature	$GLN=\frac{1}{4\pi }\sqrt{\sum_{all\;triangles}S_{i}\cdot \sum_{all\;triangles}K_{gi}^{2}S_{i}}$
MAA	Area-averaged mean curvature	$MAA=\sum_{all\;triangles}K_{mi}S_{i}/\sum_{all\;triangles}S_{i}$
MLN	L2-norm of mean curvature	$MLN=\frac{1}{4\pi }\sqrt{\sum_{all\;triangles}K_{mi}^{2}S_{i}}$

Notes—A*: cross-sectional area extracted from CAD model or using convex hull algorithm in STL points cloud; $K_{g}$: Gaussian curvature; $K_{m}$: mean curvature.

**Table 7 biomedicines-13-02914-t007:** Distribution of morphological parameters expressed as percentages, presented in total and stratified by study category.

Rank	Parameter	Frequency	Article Usage [%]			Ref.
			Total	Rupture	Growth	
1	AR	26	74.29	79.17	63.64	[20,21,22,24,27,28,29,31,32,34,35,38,39,40,41,42,43,44,45,46,47,48,50,52,53,54]
2	SR	24	68.57	75.00	54.55	[20,21,22,25,28,29,31,32,34,35,38,39,40,41,43,44,45,46,47,48,49,52,53,54]
3	Volume	15	42.86	45.83	36.36	[20,21,22,24,25,27,28,29,34,38,42,47,49,52,53]
4	NSI	14	40.00	37.50	45.45	[20,34,35,39,41,42,43,44,45,46,47,49,53,54]
5	BF	12	34.29	33.33	36.36	[31,32,34,35,39,41,42,43,44,49,53,54]
5	UI	12	34.29	29.17	45.45	[20,24,34,35,39,41,43,44,45,46,49,54]
6	Surface Area	9	25.71	29.17	18.18	[20,21,22,29,34,38,42,52,53]
7	Neck Area	7	20.00	20.83	18.18	[21,24,29,35,42,49,53]
7	HWR	7	20.00	25.00	9.09	[31,32,34,35,41,49,53]
8	VOR	6	17.14	4.17	45.45	[21,39,43,44,49,53]
8	EI	6	17.14	16.67	18.18	[20,41,45,46,49,53]
9	GAA	5	14.29	4.17	36.36	[39,43,44,49,53]
10	CR	4	11.43	4.17	27.27	[39,43,44,53]
11	IPR	2	5.71	4.17	9.09	[49,53]
11	BL	2	5.71	4.17	9.09	[49,53]
12	GLN	1	2.86	4.17	0.00	[53]
12	MAA	1	2.86	4.17	0.00	[53]
12	MLN	1	2.86	4.17	0.00	[53]

Abbreviations—AR: aspect ratio; SR: size ratio; NSI: nonsphericity index; BF: bottleneck factor; UI: undulation index; HWR: height-to-width ratio; VOR: volume-to-ostium area ratio; EI: ellipticity index; GAA: area-averaged Gaussian curvature; CR: convexity ratio; IPR: isoperimetric ratio; BL: bulge location; GLN: L2-norm of Gaussian curvature; MAA: area-averaged mean curvature; MLN: L2-norm of mean curvature.

**Table 8 biomedicines-13-02914-t008:** Distribution of hemodynamic parameters expressed as percentages, presented in total and stratified by study category.

Rank	Parameter	Frequency	Article Usage [%]			Ref.
			Total	Rupture	Growth	
1	OSI	32	91.43	95.83	81.82	[20,21,22,23,24,25,26,28,29,30,31,32,33,34,35,36,37,38,39,40,41,42,43,44,45,46,47,48,50,52,53,54]
2	TAWSS	25	71.43	66.67	81.82	[20,21,22,23,25,26,27,28,30,32,33,34,35,36,38,39,41,42,43,44,47,50,52,53,54]
3	LSAR	21	60.00	66.67	45.45	[21,22,24,26,29,30,31,33,34,35,36,38,39,40,41,42,44,46,48,50,53]
4	NWSS	18	51.43	58.33	36.36	[21,24,30,31,32,34,35,36,37,39,40,41,44,45,46,48,52,53]
5	RRT	16	45.71	50.00	36.36	[20,21,23,25,28,31,34,35,36,40,41,42,43,45,47,54]
6	VE	10	28.57	20.83	45.45	[21,22,25,30,38,39,43,44,50,53]
7	WSSG	8	22.86	16.67	36.36	[21,23,34,35,36,37,40,43]
7	ICI	8	22.86	20.83	27.27	[24,30,39,44,47,50,53,54]
8	Q	7	20.00	12.50	36.36	[27,30,37,39,44,52,53]
8	SCI	7	20.00	16.67	27.27	[24,30,39,44,50,53,54]
9	VD	4	11.43	8.33	18.18	[30,39,44,53]
9	Pressure	4	11.43	16.67	0.00	[22,25,34,38]
9	Podent	4	11.43	8.33	18.18	[30,39,44,53]
9	Corelen	4	11.43	4.17	27.27	[30,39,44,50]
9	NP	4	11.43	16.67	0.00	[31,34,35,40]
10	Flow complexity	3	8.57	8.33	9.09	[20,22,48]
10	SR	3	8.57	4.17	18.18	[43,50,53]
10	CHP	3	8.57	12.50	0.00	[35,36,40]
10	nCrPoints	3	8.57	4.17	18.18	[30,39,44]
10	VO	3	8.57	4.17	18.18	[43,50,53]
10	Flow stability	3	8.57	8.33	9.09	[20,41,48]

Abbreviations—OSI: oscillatory shear index; TAWSS: time-averaged wall shear stress; LSAR: low shear area ratio; NWSS: normalized wall shear stress; RRT: relative residence time; VE: mean velocity; WSSG: wall shear stress gradient; ICI: inflow concentration index; Q: mean inflow rate into aneurysm; SCI: shear concentration index; VD: mean viscous dissipation; Pressure: average aneurysm pressure; Podent: proper orthogonal decomposition entropy; Corelen: vortex core line length; NP: normalized pressure; SR: mean shear rate; CHP: combined hemodynamic parameter; nCrPoints: nr of WSS critical points; VO: mean vorticity.

**Table 9 biomedicines-13-02914-t009:** Percentage distribution of statistical metrics used in hemodynamic parameters related to wall shear stress.

Parameter	Metric	Frequency	Article Usage [%]			Ref.
			Total	Rupture	Growth	
OSI	Min	8	22.86	29.17	9.09	[22,24,26,35,36,38,40,47]
	Mean	28	80.00	79.17	81.82	[20,21,22,23,24,26,29,30,32,33,35,36,37,38,39,40,41,42,44,45,46,47,48,50,52,53,54]
	Max	15	42.86	37.50	54.55	[20,22,24,26,30,35,36,38,39,40,43,44,47,50,53]
	Median	4	11.43	16.67	0.00	[25,31,34,36]
TAWSS	Min	9	25.71	33.33	9.09	[22,26,28,35,36,38,42,47,53]
	Mean	25	71.43	66.67	81.82	[20,21,22,23,25,26,27,28,30,32,33,34,35,36,38,39,41,42,43,44,47,50,52,53]
	Max	15	42.86	41.67	45.45	[20,22,26,30,32,34,35,36,38,39,42,44,47,50,53]
NWSS	Min	5	14.29	20.83	0.00	[24,35,36,40,48]
	Mean	18	51.43	58.33	36.36	[20,24,30,31,32,34,35,36,37,39,40,41,44,45,46,48,52,53]
	Max	10	28.57	33.33	18.18	[24,30,34,35,36,39,40,44,48,53]
RRT	Min	4	11.43	16.67	0.00	[35,36,40,47]
	Mean	16	45.71	50.00	36.36	[20,21,23,25,28,31,34,35,36,40,41,42,43,45,47,54]
	Max	5	14.29	16.67	9.09	[20,35,36,40,47]
WSSG	Min	3	8.57	12.50	0.00	[35,36,40]
	Mean	8	22.86	20.83	27.27	[21,23,34,35,36,37,40,43]
	Max	3	8.57	12.50	0.00	[35,36,40]
CHP	Min	3	8.57	12.50	0.00	[35,36,40]
	Mean	3	8.57	12.50	0.00	[35,36,40]
	Max	3	8.57	12.50	0.00	[35,36,40]

Abbreviations—OSI: oscillatory shear index; TAWSS: time-averaged wall shear stress; NWSS: normalized wall shear stress; RRT: relative residence time; WSSG: wall shear stress gradient; CHP: combined hemodynamic parameter.

**Table 10 biomedicines-13-02914-t010:** Percentage distribution of criteria used to define the low shear area ratio range.

LSAR Threshold	Percentage of Use [%]	Ref.
One standard deviation below the mean WSS of the parent artery	22.86	[24,26,30,33,39,44,50,53,54]
Below 10% of the average WSS of the parent artery	20.00	[31,34,36,41,42,46,48,54]
Below 0.4 Pa	2.86	[29]
NI	14.29	[21,22,35,38,40,54]
NA	40.00	[20,23,25,27,30,32,37,45,47,50,51,52,54]

Abbreviations—NI: not informed; NA: not applied; WSS: wall shear stress.

## Data Availability

The original contributions presented in the study are included in the article. Further inquiries can be directed to the corresponding author.

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
