# Peer review of "Computational Fluid Dynamics Approaches for Analyzing Rupture and Growth of Intracranial Aneurysms: A Systematic Review"

_biomedicines, 2025, doi:10.3390/biomedicines13122914_

Round 1

Reviewer 1 Report

Comments and Suggestions for Authors

This review (manuscript number: biomedicines-3894032) uses rigorous methods (PRISMA and ROBINS-I) to identify consensus CFD parameters (OSI, TAWSS, AR, and SR) for intracranial aneurysm risk assessment. However, the manuscript's presentation needs improvement to maximize its impact. In some sections, the paper reads more like a bibliometric analysis than a traditional review. A final revision should transform passive recommendations into proactive directives for standardization and validation.

The following points summarize the required revisions:

  1. The abstract needs refinement by first stating that the hemodynamic metrics indicate low or oscillatory shear to connect them more clearly to pathogenesis. Next, briefly clarify that the morphological parameters AR and SR reflect aneurysm shape and relative size. The conclusion should be strengthened by urging a call to "must urgently standardize computational frameworks." The text is currently too dense and could be tightened by reducing redundancy (like repeated descriptions of metrics and the phrase "assessing the risk of rupture and growth"), streamlining phrasing, and standardizing style (e.g., "rupture and growth"). Finally, the abstract needs to briefly include the clinical significance and the relative frequency of parameter use.
  2. The following points need to be addressed in the introduction to enhance scientific clarity and credibility.

 The problem statement and knowledge gap are not clearly defined; the main text provides an overview of aneurysm risk and the use of CFD but does not clearly identify the unresolved questions or precise gaps in the literature, thereby reducing the need for the study. The literature background reads more like a sequential enumeration of previous studies than a critical review; grouping studies by topic to highlight methodological strengths, weaknesses, or inconsistencies, while ensuring that all background details are directly related to the study objectives, would help improve the study's focus. The study objective is placed at the end of the article but is phrased passively and buried within a longer paragraph; it should be stated more clearly and proactively as the final sentence, serving as a logical conclusion to the gap analysis and literature review.

  1. In the Materials and Methods section, the search strategy was inconsistently defined. The paper stated that the search covered the five years leading up to 2024, but the PubMed search query used the format "2019/01/01[date - publication] : "3000[date - publication]," which refers to future dates and is technically incorrect. The correct Scopus search query is "PUBYEAR >2018 AND PUBYEAR<2025," covering the years 2019 to 2024. The PubMed search query should be revised to ensure consistency and accuracy.
  2. Under section 2.1 Search strategy, the database selection is limited to PubMed and Scopus. Although these are major sources, the omission of other relevant databases such as Web of Science and Embase may restrict the comprehensiveness of the review. This limitation should be explicitly acknowledged in the Discussion.
  3. The PubMed search strings use only free-text [Title/Abstract], yet the text mentions DeCS/MeSH. Clarify whether MeSH terms were actually used or remove the reference.
  4. The PICO framework overlaps and creates confusion. By defining C as "studies analyzing different hemodynamic parameters or aneurysm characteristics... without prior stent treatment," the paper is incorrectly using this element to restate an exclusion criterion (line 134: "(i) analysis in aneurysm treated with Stent/Flow Diverter/Web").
  5. The authors should briefly explain why ROBINS-I is suitable for pure modeling or diagnostic studies, for example noting its utility in assessing confounding and missing data in retrospective CFD analyses.
  6. The discussion of aneurysmal blebs (lines 541–551) presents two potentially conflicting findings (low WSS combined with high OSI, versus high and heterogeneous WSS near the inflow). The review should attempt to synthesize or categorize these findings according to aneurysm type or location or simply point out that the literature is currently inconsistent regarding the precise hemodynamic triggers of aneurysmal bleb formation.
  7. The reviewed studies revealed insufficient reporting of mesh quality, with only 22.86% of studies performing mesh independence checks and the majority of studies failing to provide any mesh quality metrics. This omission severely compromises the reliability of all WSS-derived parameters, including OSI and TAWSS, and should be highlighted as a major methodological flaw, if possible, ideally in a dedicated paragraph.
  8. The Conclusion must integrate the review's significance, ensure all key findings (especially Size Ratio, SR) are included in the list of discriminators, and articulate specific, practical recommendations for standardization.

Author Response

Reviewer 1

This review (manuscript number: biomedicines-3894032) uses rigorous methods (PRISMA and ROBINS-I) to identify consensus CFD parameters (OSI, TAWSS, AR, and SR) for intracranial aneurysm risk assessment. However, the manuscript's presentation needs improvement to maximize its impact. In some sections, the paper reads more like a bibliometric analysis than a traditional review. A final revision should transform passive recommendations into proactive directives for standardization and validation.

The following points summarize the required revisions:

  1. The abstract needs refinement by first stating that the hemodynamic metrics indicate low or oscillatory shear to connect them more clearly to pathogenesis. Next, briefly clarify that the morphological parameters AR and SR reflect aneurysm shape and relative size. The conclusion should be strengthened by urging a call to "must urgently standardize computational frameworks." The text is currently too dense and could be tightened by reducing redundancy (like repeated descriptions of metrics and the phrase "assessing the risk of rupture and growth"), streamlining phrasing, and standardizing style (e.g., "rupture and growth"). Finally, the abstract needs to briefly include the clinical significance and the relative frequency of parameter use.

Answer: We appreciate the reviewer’s valuable comment. Based on this feedback, we have revised the abstract to more clearly describe the relationship between hemodynamic parameters and aneurysm pathogenesis, clarified the definitions and relative frequencies of key parameters, and emphasized the urgent need to standardize computational frameworks, parameter definitions, and boundary conditions to improve the consistency, comparability, and clinical applicability of CFD in aneurysm risk assessment.

  1. The following points need to be addressed in the introduction to enhance scientific clarity and credibility: The problem statement and knowledge gap are not clearly defined; the main text provides an overview of aneurysm risk and the use of CFD but does not clearly identify the unresolved questions or precise gaps in the literature, thereby reducing the need for the study. The literature background reads more like a sequential enumeration of previous studies than a critical review; grouping studies by topic to highlight methodological strengths, weaknesses, or inconsistencies, while ensuring that all background details are directly related to the study objectives, would help improve the study's focus. The study objective is placed at the end of the article but is phrased passively and buried within a longer paragraph; it should be stated more clearly and proactively as the final sentence, serving as a logical conclusion to the gap analysis and literature review.

Answer: We appreciate the reviewer’s comment aimed at improving the quality of our manuscript. Accordingly, we have revised the Introduction to better define the research problem and knowledge gap, clarify the motivation for the study, and explicitly state the objective and rationale for conducting this systematic review.

  1. In the Materials and Methods section, the search strategy was inconsistently defined. The paper stated that the search covered the five years leading up to 2024, but the PubMed search query used the format "2019/01/01[date - publication] : "3000[date - publication]," which refers to future dates and is technically incorrect. The correct Scopus search query is "PUBYEAR >2018 AND PUBYEAR<2025," covering the years 2019 to 2024. The PubMed search query should be revised to ensure consistency and accuracy.

Answer: We appreciate your observation. We have corrected the search strategy by specifying the reference date used to delimit the search period. The revised PubMed search strategy is as follows:

((((("Computational Fluid Dynamics"[Title/Abstract]) OR (CFD[Title/Abstract])) AND ((("Intracranial Aneurysm"[Title/Abstract]) OR ("Cerebral Aneurysm"[Title/Abstract])) OR ("Brain Aneurysm"[Title/Abstract]))) AND (English[Language])) NOT (review[Publication Type])) AND (("2019/01/01"[Date - Publication] : "2024/08/14"[Date - Publication]))

  1. Under section 2.1 Search strategy, the database selection is limited to PubMed and Scopus. Although these are major sources, the omission of other relevant databases such as Web of Science and Embase may restrict the comprehensiveness of the review. This limitation should be explicitly acknowledged in the Discussion.

Answer: We agree with you and have updated the Search Strategy subsection in the Materials and Methods section, incorporating the complete search strategy and a new Figure 1 illustrating the systematic review process, which now includes both the Web of Science and Embase databases. A total of 284 records were identified in Web of Science and 228 in Embase. In Web of Science, one record corresponded to a book chapter and two to meeting abstracts, while the remaining 281 were duplicates of studies already retrieved from Scopus and PubMed. All records identified in Embase were also duplicates of articles already present in PubMed, Scopus, or Web of Science, and therefore did not alter the final selection of studies included in the review. The detailed search strategy is presented below and has also been integrated into the manuscript text.

  1. The PubMed search strings use only free-text [Title/Abstract], yet the text mentions DeCS/MeSH. Clarify whether MeSH terms were actually used or remove the reference.

Answer:  We appreciate the observation. Indeed, Mesh terms were not used, so we have corrected this by removing this mention.

  1. The PICO framework overlaps and creates confusion. By defining C as "studies analyzing different hemodynamic parameters or aneurysm characteristics... without prior stent treatment," the paper is incorrectly using this element to restate an exclusion criterion (line 134: "(i) analysis in aneurysm treated with Stent/Flow Diverter/Web").

Answer:  We sincerely appreciate the reviewer’s valuable comment and the opportunity to clarify this point. The necessary revisions have been made to improve the clarity of the text. The PICO framework is a mandatory element required by PROSPERO; however, this systematic review does not address clinical interventions, but rather focuses on engineering studies involving computational modeling and simulations. Consequently, there are no true “interventions,” “comparators,” or “outcomes” in the conventional clinical sense. To comply with PROSPERO requirements, the PICO structure was therefore adapted accordingly. Additionally, to avoid any potential confusion or misinterpretation regarding to restate the exclusion criterion, we removed the reference to “without prior stent treatment.” In the PICO section.

  1. The authors should briefly explain why ROBINS-I is suitable for pure modeling or diagnostic studies, for example noting its utility in assessing confounding and missing data in retrospective CFD analyses.

Answer:  Following the PRISMA methodology, bias assessment is a mandatory component, with several established tools available—such as ROBINS-I, the Newcastle–Ottawa Scale, and PROBAST—each tailored to specific types of studies commonly found in biological and medical research, including randomized clinical trials, case-control, and cohort studies. However, none of these categories adequately encompass CFD studies or other engineering-based modeling investigations. For this reason, ROBINS-I was selected, as it provides greater flexibility to adapt its domains to the identification of potential confounders or sources of bias in CFD analyses. Among the available tools, ROBINS-I allows most questions to be meaningfully adapted, whereas the others would largely remain “not applicable” due to the fundamental differences in study design and methodology. However, it was an adaptation made by the authors themselves in order to meet the mandatory requirements of PRISMA.

We have amended the text to provide a brief explanation of the general inapplicability of these usual methods, as well as clarifying the adaptation for our case.

  1. The discussion of aneurysmal blebs (lines 541–551) presents two potentially conflicting findings (low WSS combined with high OSI, versus high and heterogeneous WSS near the inflow). The review should attempt to synthesize or categorize these findings according to aneurysm type or location or simply point out that the literature is currently inconsistent regarding the precise hemodynamic triggers of aneurysmal bleb formation.

Answer:  We appreciate the reviewer’s comment, and we have revised the text to enhance clarity and understanding. An additional explanation has been included at the end of the same paragraph (lines 541–551). It is important to note that these findings are not necessarily contradictory, as both low and high WSS represent conditions outside the normal or physiological range. There is a possibility that both may contribute to bleb formation through distinct but complementary mechanisms. However, there is still no consensus in the literature regarding whether both are key factors acting together, or if one predominates while the other is secondary or whether other simultaneous or more dominant triggers may also be involved.

  1. The reviewed studies revealed insufficient reporting of mesh quality, with only 22.86% of studies performing mesh independence checks and the majority of studies failing to provide any mesh quality metrics. This omission severely compromises the reliability of all WSS-derived parameters, including OSI and TAWSS, and should be highlighted as a major methodological flaw, if possible, ideally in a dedicated paragraph.

Answer: We appreciate the suggestion, and we agree with this point. We have adapted the discussion to give more emphasis to this topic in a dedicated paragraph.

  1. The Conclusion must integrate the review's significance, ensure all key findings (especially Size Ratio, SR) are included in the list of discriminators, and articulate specific, practical recommendations for standardization.

Answer: The conclusion section has been rewritten to enhance clarity and cohesion. It now integrates the overall significance of the review, explicitly includes the Size Ratio (SR) among the key discriminators while ensuring that all other major findings are properly addressed, and provides specific and practical recommendations to support standardization in future studies.

Reviewer 2 Report

Comments and Suggestions for Authors

We read with interest the article by Loly et al., where the authors present a comprehensive and methodically sound systematic review examining the application of computational fluid dynamics (CFD) in studying intracranial aneurysm (IA) rupture and growth. The authors effectively compile methodological data across 35 studies from 2019–2024, emphasizing hemodynamic and morphological parameters such as wall shear stress (WSS), oscillatory shear index (OSI), aspect ratio (AR), and size ratio (SR). The topic is timely, and the paper fills a gap by focusing on methodological harmonization within CFD research, which is often fragmented.

There are strengths and novelty in the study related to the following points:

The study clearly follows PRISMA guidelines, outlining database searches, inclusion/exclusion criteria, and data extraction methods. The registration on PROSPERO enhances transparency and credibility. The extensive use of tables and figures—such as distributions of mesh types, solver preferences, and hemodynamic parameters—facilitates readability and allows technical comparison across studies. The PRISMA flowchart and quantitative synthesis is  well presented. The detailed review of segmentation, meshing, and solver software (e.g., MIMICS, Ansys Fluent, OpenFOAM) gives practical insight into current CFD workflows in aneurysm modeling.

However, there are some weaknesses:

1- The review is largely descriptive—cataloging tools and metrics—without deeply critiquing why certain methods or parameters yield more reliable predictions of rupture. The lack of comparative or meta-analytic quantification weakens interpretive depth. Some important CFD variables (e.g., mesh independence validation, wall compliance, patient-specific inflow profiles) are mentioned but not comparatively analyzed in relation to outcome reliability. This limits the ability to infer best practices. the manuscript do  not attempt pooled effect-size comparisons or sensitivity analyses. Even simple frequency-weighted correlations between AR, OSI, or rupture outcomes could have added analytical rigor. Finalyy, the paper does not elaborate on available validation studies, benchmarking initiatives, or reproducibility frameworks that could support translation into clinical decision-making.

Minor Comments:

their are  sentences are overly long and dense with technical terms, which could be simplified for non-engineering readers. Certain figure legends (e.g., Figures 2–4) could better contextualize significances rather than just summarize distribution.

Comments on the Quality of English Language

Minor corrections,

long sentences need to be shortened.

Author Response

Reviewer 2

We read with interest the article by Loly et al., where the authors present a comprehensive and methodically sound systematic review examining the application of computational fluid dynamics (CFD) in studying intracranial aneurysm (IA) rupture and growth. The authors effectively compile methodological data across 35 studies from 2019–2024, emphasizing hemodynamic and morphological parameters such as wall shear stress (WSS), oscillatory shear index (OSI), aspect ratio (AR), and size ratio (SR). The topic is timely, and the paper fills a gap by focusing on methodological harmonization within CFD research, which is often fragmented.

There are strengths and novelty in the study related to the following points:

The study clearly follows PRISMA guidelines, outlining database searches, inclusion/exclusion criteria, and data extraction methods. The registration on PROSPERO enhances transparency and credibility. The extensive use of tables and figures—such as distributions of mesh types, solver preferences, and hemodynamic parameters—facilitates readability and allows technical comparison across studies. The PRISMA flowchart and quantitative synthesis is  well presented. The detailed review of segmentation, meshing, and solver software (e.g., MIMICS, Ansys Fluent, OpenFOAM) gives practical insight into current CFD workflows in aneurysm modeling.

However, there are some weaknesses:

  • The review is largely descriptive—cataloging tools and metrics—without deeply critiquing whycertain methods or parameters yield more reliable predictions of rupture. The lack of comparative or meta-analytic quantification weakens interpretive depth. Some important CFD variables (e.g., mesh independence validation, wall compliance, patient-specific inflow profiles) are mentioned but not comparatively analyzed in relation to outcome reliability. This limits the ability to infer best practices. the manuscript do not attempt pooled effect-size comparisons or sensitivity analyses. Even simple frequency-weighted correlations between AR, OSI, or rupture outcomes could have added analytical rigor. Finalyy, the paper does not elaborate on available validation studies, benchmarking initiatives, or reproducibility frameworks that could support translation into clinical decision-making.

Answer: We sincerely appreciate the reviewer's thoughtful and constructive comments, which have undoubtedly strengthened the scientific rigor and depth of our manuscript. We have carefully considered each suggestion and implemented revisions accordingly.

Regarding sensitivity analysis, effect size calculations, and meta-analytic quantification, we would like to respectfully clarify that the primary objective of this work was to conduct a comprehensive descriptive systematic review rather than a meta-analysis. This methodological choice was deliberately made due to the substantial heterogeneity observed across the included studies—encompassing variations in computational parameters (mesh resolution, time step size, solver selection), diverse modeling approaches (rheological assumptions, boundary condition strategies, turbulence models), anatomical diversity (patient-specific versus idealized geometries), and inconsistent reporting of hemodynamic outcomes (wall shear stress metrics, velocity profiles, pressure distributions). This marked heterogeneity, while reflecting the evolving and multifaceted nature of cerebrovascular CFD research, precluded meaningful quantitative synthesis or pooled statistical analysis that would be characteristic of a traditional meta-analysis.

Nevertheless, we recognize the value of providing deeper analytical insights beyond descriptive summary. In response to the reviewer's valuable suggestions, we have substantially expanded both the Results and Discussion sections to offer more nuanced interpretations of how specific methodological choices influence computational outcomes and clinical relevance. These enhancements include critical evaluation of numerical error control strategies and their impact on solution accuracy, assessment of how software selection (commercial versus open-source platforms) affects reproducibility and accessibility, detailed analysis of the influence of patient-specific versus generic boundary conditions on hemodynamic fidelity, and exploration of mesh quality metrics and their relevance of results reliability and transparency.

  1. their are  sentences are overly long and dense with technical terms, which could be simplified for non-engineering readers. Certain figure legends (e.g., Figures 2–4) could better contextualize significances rather than just summarize distribution.

Answer: We appreciate the reviewer’s thoughtful comments. In response, we have revised the captions of the referenced figures and edited the text to make it more concise and accessible to readers without an engineering background.

Round 2

Reviewer 1 Report

Comments and Suggestions for Authors

The revised draft is acceptable from my side.